# IL-21 and IFNα therapy rescues terminally differentiated NK cells and limits SIV reservoir in ART-treated macaques

Justin Harper [1,11], Nicolas Huot [2,11], Luca Micci[1], Gregory Tharp [3], Colin King[1], Philippe Rascle[2,4],
Neeta Shenvi[5], Hong Wang[1], Cristin Galardi[6,7], Amit A. Upadhyay[3], Francois Villinger[8], Jeffrey Lifson[9],
Guido Silvestri[1,10], Kirk Easley [5], Beatrice Jacquelin [2], Steven Bosinger [1,3,10],
Michaela Müller-Trutwin [2,11] & Mirko Paiardini [1,10,11 ✉]

Unlike HIV infection, which progresses to AIDS absent suppressive anti-retroviral therapy, nonpathogenic infections in natural hosts, such African green monkeys, are characterized by a lack of gut microbial translocation and robust secondary lymphoid natural killer cell responses resulting in an absence of chronic inflammation and limited SIV dissemination in lymph node B-cell follicles. Here we report, using the pathogenic model of antiretroviral therapy-treated, SIV-infected rhesus macaques that sequential interleukin-21 and interferon alpha therapy generate terminally differentiated blood natural killer cells (NKG2a/c$^{low}$CD16$^+$) with potent human leukocyte antigen-E-restricted activity in response to SIV envelope peptides. This is in contrast to control macaques, where less differentiated, interferon gamma-producing natural killer cells predominate. The frequency and activity of terminally differentiated NKG2a/c$^{low}$CD16$^+$ natural killer cells correlates with a reduction of replication-competent SIV in lymph node during antiretroviral therapy and time to viral rebound following analytical treatment interruption. These data demonstrate that African green monkey-like natural killer cell differentiation profiles can be rescued in rhesus macaques to promote viral clearance in tissues.

[1] Division of Microbiology and Immunology, Yerkes National Primate Research Center, Emory University, Atlanta, GA, USA. [2] Institut Pasteur, Unité HIV, Inflammation et Persistance, Paris, France. [3] Nonhuman Primate Genomics Core, Yerkes National Primate Research Center, Emory University, Atlanta, GA, USA. [4] Université Paris Diderot, Sorbonne Paris Cité, Paris, France. [5] Department of Biostatistics and Bioinformatics, Rollins School of Public Health, Emory University, Atlanta, GA, USA. [6] UNC HIV Cure Center and Department of Medicine, University of North Carolina at Chapel Hill, Chapel Hill, NC, USA. [7] HIV Discovery, ViiV Healthcare, Research Triangle Park, NC, USA. [8] Department of Biology, New Iberia Research Center, University of Louisiana at Lafayette, New Iberia, LA, USA. [9] AIDS and Cancer Virus Program, Frederick National Laboratory for Cancer Research, Frederick, MD, USA. [10] Department of Pathology and Laboratory Medicine, Emory University School of Medicine, Atlanta, GA, USA. [11] These authors contributed equally: Justin Harper, Nicolas Huot, Michaela Müller-Trutwin, Mirko Paiardini. ✉email: mirko.paiardini@emory.edu

Natural killer (NK) cells are "licensed" with functional competence following education with self-major histo-compatibility complex (MHC) class I molecules[1]. In particular human leukocyte antigen (HLA)-E, the ligand for the inhibitory CD94/NKG2a receptor[2–4], is positively regulated by HLA-A expression and inhibits NK cell-mediated lysis[5,6]. In a companion manuscript[7], Huot et al. define NK cell differentiation states based on their education via NKG2a and expression of CD16 (FcγRIII), an activating Fc receptor that mediates antibody-dependent cell-mediated cytotoxicity (ADCC)[8], and demonstrate that nonpathogenic SIVagm infection in African green monkeys (AGMs) imprints the maturation of NK cells inducing terminally differentiated NKG2a$^{low}$CD16$^+$ NK cells, which express high levels of interleukin (IL)−21R. It has been previously demonstrated that NK cell functionality is responsive to immunotherapies with IL-21 and interferon alpha (IFNα). For example, ex vivo IL-21 treatment expanded CD16$^+$ NK cells[9], antagonized the IL-15-dependent expansion of resting NK cells[10], and reverses hypo-responsiveness via the STAT1 and PI3K-AKT-FOXO1 pathways[11]. Likewise, ex vivo IFNα therapy upregulates IL-15-mediated NK cell cytotoxicity[12], including CD107a degra-nulation and ADCC activity[13,14], and downregulates IL-21R expression[15]; furthermore, in vivo IFNα-induced NK cell cyto-toxicity correlates with reductions in HIV-DNA during anti-retroviral therapy (ART)[16]. In SIVagm infection, systemic IL-15 was associated with NK cell proliferation in lymph node (LN), while systemic IFNα correlated with NK cell cytotoxicity in LN[17]. Given previous findings on a role of IL-21 and IFNα in regulating NK cell function, and that NKG2a$^{low}$CD16$^+$ NK cells are gen-erated while expressing high levels of IL-21R in nonpathogenic SIV infection, we sought to determine whether immunotherapy with IL-21 and IFNα rescues AGM-like profiles of NK cell maturation and activity in SIV-infected rhesus macaques (RMs).

## Results

### IL-21 and rIFNα immunotherapies are biologically active in SIV-infected, ART-treated RMs

Sixteen RMs were intrave-nously (i.v.) infected with SIVmac$_{239}$ and at day (d) 35 post-infection (p.i.) initiated triple formulation ART[18], which was maintained for 13 months (Fig. 1a and Supplementary Table 1). Prior to ART initiation, the RMs RPk11 and RNa12 mimicked pre-established virologic and immunologic features of controllers; hence, both were not assigned to an experimental group, but excluded from analyses and followed as a part of a study aimed at characterizing post-treatment viral control. Among the remaining 14 RMs, 9 were administered rhesus rIL-21-IgFc (IL-21) at d42 and d189 p.i. in two cycles of four doses given once per week followed by weekly rhesus IFNα-IgFc (rIFNα) starting at d323 (3 doses) and d383 p.i. (2 doses; i.e., ART + IL-21 + rIFNα, cyto-kine-treated). The cytokine-treated RM 172_10 was euthanized at d66 p.i. due to AIDS-defining conditions. Five RMs served as cytokine treatment-naive, ART-only controls (i.e., controls). ART was withdrawn at d402 p.i. and, given attenuation in IFN sig-naling upon sustained therapy[19], cytokine treatment-experienced RMs were transitioned to human PEGylated-IFNα (PEG-IFNα; 7 doses, once every 6–8 days, subcutaneous (s.c.), 7 μg/kg; i.e., cytokine-treated) followed by necropsy in 6 months. IL-21 and PEG-IFNα sequential therapies were well tolerated without clin-ical complications as anticipated based on prior monotherapy administration in SIV-infected RMs[20,21]. Plasma viral loads amid ART revealed no treatment-related impact on the kinetics of viral suppression or rate of viral reactivation (Fig. 1b, c).

To confirm biological activity, we sought to recapitulate observations that IL-21 attenuates residual T-cell immune activation and improves mucosal immunity during ART[20,22]. In cytokine-treated RMs, ART with IL-21 treatment was superior, as compared to ART-only controls, in rapidly and significantly reducing immune activation (HLA-DR$^+$CD38$^+$) in memory CD4$^+$ T-cells from peripheral blood mononuclear cells (PBMCs; Fig. 1d with representative stains and gating strategy in Fig. 1e and Supplementary Fig. 1a, respectively). A similar early reduction following IL-21 treatment was found in rectal biopsy (RB) for the levels of immune activation in memory CD4$^+$ T-cells (Supplementary Fig. 2a) and proliferation (Ki-67$^+$) in CD8$^+$ T-cells (Supplementary Fig. 2j); however, changes in RB did not sustain for long term and treatment did not substantially impact activation or proliferation in CD4$^+$ and CD8$^+$ T-cells from LN (Supplementary Fig. 2). In addition, IL-21 therapy significantly enhanced Th17/Th22 functionality based on the expression of IL-2 and TNF-α (Supplementary Fig. 3). The efficacy of rIFNα amid long-term ART was confirmed by the upregulation of IFN-stimulated genes (ISGs) in PBMCs at 2 h post-treatment relative to control RMs (Fig. 1f), which utilized six rIFNα-treated and two of five control RMs that were not described in this manuscript. In a different historical cohort[23], we also confirmed that ISGs are induced by pathogenic SIVmac infection and are significantly reduced, but not fully normalized, by ART (Supplementary Fig. 4), as is observed in natural hosts[24,25]. Cytokine treatment reduced the frequency of HLA-E$^+$CD4$^+$ T-cells (Fig. 1g, representative gating strategy in Supplementary Fig. 1b); however, it did not impact the frequency of NKG2a/c$^+$ CD8$^+$ T-cells (Fig. 1h, representative gating strategy in Supplementary Fig. 1c) nor did it enhance T-cell responses whether by T-bet expression, which regulates Th1 cytokine expression[26], or by IFN-γ ELISpot following stimulation with SIV-Gag or -envelope (Env) peptides (Supplementary Fig. 5; T-bet gating strategy given in Supplementary Fig. 1d). We then analyzed if the cytokine therapy enhanced NK cell activity by exposing them to MHC-I-deficient target cells with induced HLA-E loaded with SIVmac$_{239/251}$ Env peptides. By measuring levels of surface CD107a expression, we calculated the SIV-Env-specific, HLA-E-restricted NK cell activity (raw data of CD107a expression by co-culture condition are given in Supplementary Fig. 6–c; representative stains shown in Supplemen-tary Fig. 6d–f; see Eq. 1). IL-21 administration led to a significant retention, which was sustained following rIFNα administration, of the Env-specific NK cell activity independent of viremia (Fig. 1i), indicating that the designed immunotherapy impacted NK cell imprinting while CD8$^+$ T-cell responses remained tepid.

### Cytokine therapy reduces replication competent virus in lym-phoid tissue, which is uniquely correlated with Env-specific NK cell activity

As cytokine therapy enhanced Env-specific NK cell activity, we sought to determine the impact on viral persistence amid ongoing ART. Independent of ART-mediated viral sup-pression, cytokine therapy failed to reduce the content of total cell-associated SIV-RNA (Fig. 2a–c) or -DNA (Fig. 2d–f) as deter-mined by quantitative reverse transcription polymerase chain reaction (qRT-PCR) in bulk PBMCs or LN when compared to controls; yet, cell-associated SIV-DNA in RB was lower in cytokine-treated animals as compared to controls ($p = 0.0576$ at d374 p.i.; Fig. 2f). Moreover, in cytokine-treated RMs, IL-21 therapy (d217 p.i.) significantly decreased the frequency of LN CD4$^+$ cells harboring replication competent virus relative to controls, as determined by quantitative viral outgrowth assay (QVOA) (Fig. 2g). In three out of four controls, viral suppression due to ongoing ART (d217 to d374 p.i.) resulted in a non-significant reduction of replication competent virus. In cytokine-treated RMs, subsequent rIFNα therapy did not further decline the replication competent viral content, which remained significantly lower than controls also at d374 p.i. (Fig. 2g). Consistent with previous observations[20], the IL-21 impact on replication compe-tent virus was likely unrelated to SIV-specific T-cell responses

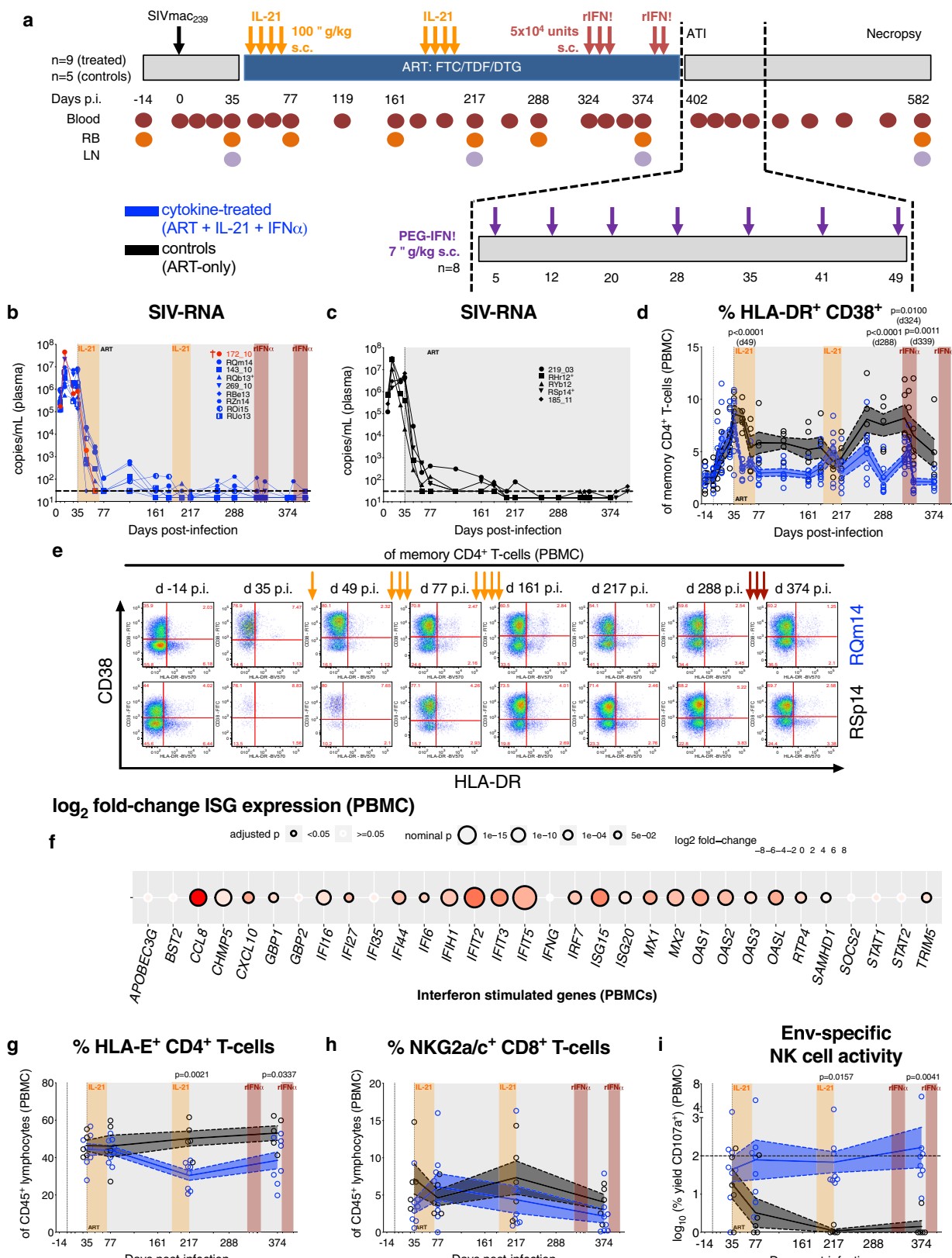

(Supplementary Fig. 5d, e). To further understand the effect on the viral reservoir, measures of SIV reservoir content were correlated against measures of T-cell immune activation, T-cell proliferation, HLA-E⁺ CD4⁺ T-cells, NKG2a/c⁺ CD8⁺ T-cells, and Env-specific NK cell activity (Fig. 2h). The content of cell-associated SIV-RNA and -DNA, which were not impacted by therapy,

positively correlated with T-cell activation and proliferation as expected[27,28]. The frequency of LN CD4⁺ cells harboring replication competent virus was also positively correlated with T-cell activation and proliferation in blood, but not in LN. Given the enrichment of effector cells in blood as compared to lymphoid tissues[29], it is plausible that the levels of T-cell activation and

**Fig. 1 IL-21 and rIFNα immunotherapies are biologically active in SIV-infected, ART-treated RMs. a** Cartoon schematic of study design as detailed in the Results and Methods. Plasma viral loads (SIV-RNA copies/mL) were longitudinally measured by qRT-PCR with individual **b** cytokine-treated (ART + IL-21 + IFNα; blue; $n = 9$ RMs) and **c** control (ART-only; black; $n = 5$ RMs) RMs represented by distinct shapes. The dashed horizontal line represents the assay's limit of detection (30 copies/mL) with undetectable events plotted as 15 copies/mL. **d** The frequency of immune activation (HLA-DR$^+$CD38$^+$) was quantified in memory (CD95$^+$) CD4$^+$ T-cells in PBMCs and **e** representative flow cytometry plots are given at critical time points (as indicated above) for a cytokine-treated (RQm14) and control RMs (RSp14; 16 of 260 single-replicate stains). **f** The expression of IFN-stimulated genes (ISGs) was measured by RNA-seq in PBMCs at 2 h post-rIFNα and was calculated as a log$_2$ fold-change between rIFNα-treated ($n = 6$) and control RMs ($n = 5$), which is represented as a double-gradient heatmap. The size of each data point corresponds inversely to the log$_{10}$-transformed nominal $p$ value with significant ($p < 0.05$) adjusted $p$ values indicated by a black border. Using DESeq2, data were analyzed with a two-sided (95% CI) Wald test using the Benjamini–Hochberg method for multiple comparisons. By flow cytometry, the frequency of **g** HLA-E$^+$ CD4$^+$ T-cells and **h** NKG2a/c$^+$ CD8$^+$ T-cells was quantified of CD45$^+$ lymphocytes in PBMCs. **i** NK cells isolated from PBMCs were cultured alone or the presence of K562-HLA-E$^*$0101 cells either unloaded or loaded with SIVmac$_{239/251}$ Env peptides. The SIV-Env-specific, HLA-E-restricted activity was calculated as the log$_{10}$-transformed percent yield of CD107a surface expression on NK cells by flow cytometry and the horizontal dashed line represents 100% activity. **b–d**, **g–i** Treatment phases are indicated with the following background shading: IL-21 (orange), rIFNα (red), and ART (gray). **d**, **g–i** Data from individual RMs (staggered open circles) are overlaid against the mean (solid line) ± SEM (shaded region within the dashed lines): control (black; $n = 5$) and cytokine-treated (blue; $n = 8$). Data were analyzed with a **d**, **g–i** two-sided (95% CI), two-way ANOVA with Bonferroni's correction for multiple comparisons with cross-sectional comparisons relative to controls.

proliferation in blood may better represent therapy-induced immunologic and virologic changes, including in the content of replication competent virus, in LN and other, not accessed anatomical sites. Of note, the content of LN replication competent virus, but not cell-associated SIV-DNA in tissues, displayed a unique positive correlation with HLA-E$^+$ CD4$^+$ T-cell levels and a negative correlation with Env-specific NK cell activity; suggesting that enhanced NK cell functionality is an important mechanism for IL-21-mediated reduction of the replication competent viral reservoirs.

**Cytokine therapy promotes the maturation of NKG2a/c$^{low}$CD16$^+$ NK cells with enhanced ex vivo innate activity, which correlates with the content of lymphoid replication competent virus.** To better characterize the NK cell-mediated response during ART, blood NK cells were immunophenotyped for biomarkers of homing and differentiation. Cytokine therapy resulted in a non-significant increase in the frequency of total NK cells (CD45$^+$CD20$^-$CD3$^-$NKG2a/c$^+$; Fig. 3a) and did not impact homing to the B-cell follicle as gauged by CXCR5 expression (Fig. 3b), the expression of activating receptors (i.e., NKp30, NKp80, and NKp46), or cellular activation (HLA-DR) (Supplementary Fig. 7, representative gating strategy in Supplementary Fig. 1c). NK cells were divided into distinct differentiation stages (stages 0–3) based on their expression of CD16 and NKG2a/c (Fig. 3c–f; representative gating strategy and plots in Supplementary Fig. 8)[7] with the caveat that the anti-NKG2a monoclonal antibody (mAb) (clone Z199) cannot distinguish between NKG2a and NKG2c in nonhuman primates (NHPs). Cytokine therapy significantly favored the generation of the terminally differentiated NKG2a/c$^{low}$CD16$^+$ subset (Stage 3; Fig. 3f, g) with a concomitant loss of the intermediate NKG2a/c$^{high}$CD16$^+$ subset as a proportion of NK cells (Stage 2; Fig. 3e, g). Specifically, at d374 p.i. stage 3 NK cells constitute 46.3 ± 7.52% of total NK cells in cytokine-treated RMs as compared to 7.59 ± 2.24% in controls. Notably, cytokine therapy did not impact the frequency of these NK cell subsets relative to CD45$^+$ lymphocytes (Supplementary Fig. 9a–d). These data suggest that maturation was blocked in SIVmac infection, even under ART, in favor of intermediate NKG2a/c$^{high}$CD16$^+$ NK cells, while cytokine therapy allowed NK cells to attain terminal differentiation.

We next analyzed the innate activity of NK cells ex vivo. In cytokine-treated RMs, we observed a transient induction of IFN-γ expression (Fig. 3h, representative gating strategy in Supplementary Fig. 1c), but an increase in CD107a degranulation, specifically within the NKG2a/c$^{low}$CD16$^+$ subset (Fig. 3i)[30]. Although other NK cell differentiation subsets displayed variable levels of ex vivo innate

degranulation activity, cytokine therapy skewed the total ex vivo innate activity toward being dominated by stage 3 NK cells by d374 p.i. (Supplementary Fig. 9e–h). Indeed, at all measured experimental points (d77, d217, and d374), the average %CD107a$^+$ stage 3 NK cells were more than seven-fold higher in cytokine-treated than control RMs. To delineate the impact of cytokine therapy-mediated NK cell differentiation, the frequency of each differentiation subset was correlated against measures of SIV persistence, and innate and Env-specific NK cell activities in all RMs (Fig. 3j). The levels of intermediate NKG2a/c$^{high}$CD16$^+$ NK cells (stage 2) were positively correlated with replication competent virus in LN CD4$^+$ cells and negatively correlated with the Env-specific NK cell activity, in line with a lack of capacity of these cells to eliminate the persistent reservoir. In sharp contrast to the less mature NK cells, the levels of stages 3 NK cells were negatively and positively correlated with replication competent virus in LN CD4$^+$ cells and the Env-specific NK cell activity, respectively (Fig. 3j). These data imply that the terminally differentiated NKG2a/c$^{low}$CD16$^+$ NK cells (stage 3) were responsible for the Env-specific activity and possess effector responses capable of eliminating cells specifically harboring replication competent virus during ART.

**The formation and activity of NKG2a$^{low}$CD16$^+$ NK cells correlate with viral recrudescence following ATI.** To further analyze the functional relevance of NK cell differentiation in viral persistence, all RMs underwent ART analytical treatment interruption (ATI) with cytokine-treated RMs additionally receiving ongoing PEG-IFNα (Fig. 1a). Of note, PEG-IFNα therapy has previously been suggested as able to delay viral rebound when initiated prior to ATI[31,32], whereas in SIV-infected RMs, prior IL-21 monotherapy during ART is not[20]. Based on longitudinal plasma viremia following ATI (Fig. 4a, b), cytokine-treated RMs exhibited a significant delay in rebound (>200 copies/mL) both by survival curve analysis (Fig. 4c) and by day of rebound (average 22.1 ± 4.27 days versus 10.6 ± 0.98 days; Fig. 4d). Thus, cytokine treatment modulated the kinetics of plasma rebound, as a slope (d13–d20 ATI) and mean analysis (d13 ATI, Fig. 4e). Cytokine therapy did not however impact the peak or set-point viremia relative to controls (Supplementary Fig. 10a, b) or the content of cell-associated SIV-DNA or -RNA in PBMCs (Supplementary Fig. 10c, d). Transitioning treatment-experienced RMs (i.e., cytokine-treated RMs with prior rIFNα during ART) to PEG-IFNα led to a reset in ISG expression as of 24 h following the first administration (d6 ATI); however, this effect was largely lost following the fifth dose by which nearly all RMs had rebounded (d37 ATI; Fig. 5a). As with rIFNα on-ART, PEG-IFNα therapy following ATI failed to improve SIV-specific T-

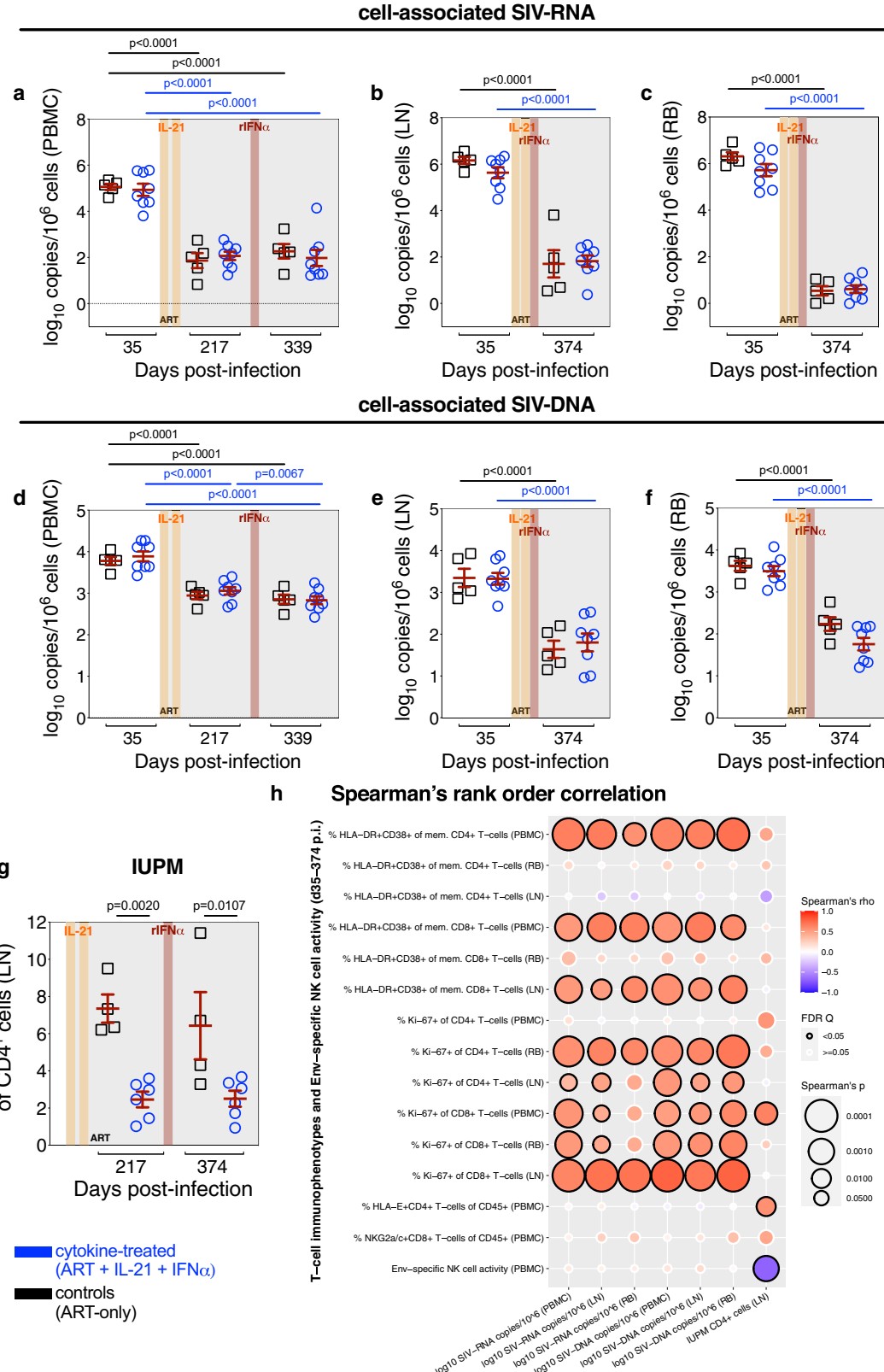

cell responses by SIV-Gag-stimulated IFN-γ ELISpot (Supplementary Fig. 10e). At d13 post ATI, the distribution of NK cell maturation subsets in PBMCs was similar to that observed during ART with higher levels of terminally differentiated (NKG2a/c^lowCD16+, stage 3) NK cells with strong, innate degranulation activity (i.e., ex vivo CD107a surface expression) in the cytokine-

treated animals in contrast to higher levels of the intermediate (NKG2a/c^highCD16+, stage 2) NK cells with weak innate degranulation activity in the controls (Fig. 5b, c). In contrast, at d13 post ATI, the Env-specific activity of bulk NK cells in cytokine-treated animals was no longer statistically significantly different from controls (Fig. 5d) and tended to converge with the levels observed in

**Fig. 2 Cytokine therapy reduces replication competent virus in lymphoid tissue, which is uniquely correlated with Env-specific NK cell activity.** The content of cell-associated SIV-RNA ($\log_{10}$ copies per $10^6$ cells) was determined by qRT-PCR in snap frozen pellets of **a** PBMCs, **b** LN, and **c** RB prior to ART initiation (d35 p.i.), following two IL-21 cycles (d217 p.i.), and after one subsequent rIFNα cycle amid ongoing ART (d339 p.i. or d374 p.i.), as was cell-associated SIV-DNA ($\log_{10}$ copies per $10^6$ cells) in **d** PBMCs, **e** LN, and **f** RB. **a–f** Data from individual RMs are overlaid against the mean ± SEM (in red): control (ART-only, black square; $n = 5$ RMs) and cytokine-treated (ART + IL-21 + IFNα, blue circle; $n = 8$ RMs). **g** From cryo-preserved, magnetically isolated LN CD4$^+$ cells, the number of infectious units per million (IUPM) cells were measured with a limiting dilution quantitative viral outgrowth assay (QVOA; 3 serial dilutions plated in triplicate) in control ($n = 4$) and cytokine-treated RMs ($n = 6$). **a–g** Treatment phases are indicated with the following background shading: IL-21 (orange), rIFNα (red), and ART (gray). Data were analyzed with two-sided (95%), two-way ANOVA with Bonferroni's correction with comparisons relative to controls and time. **h** The SIV reservoir contents in tissue (as indicated below) were correlated against phenotypes of activation (HLA-DR$^+$CD38$^+$) and proliferation (Ki-67$^+$) in CD4$^+$ and CD8$^+$ T-cells; the frequency of HLA-E$^+$ CD4$^+$ T-cells and NKG2a/c$^+$ CD8$^+$ T-cells; and the Env-specific NK cell activity (as indicated at left) in all RMs ($n = 13$; days 35, 217, and 339/374 p.i. as matched data are available). Per each correlation the two-tailed (95% CI) Spearman's rank correlation coefficient (rho) is represented as a double-gradient heatmap and the size of each data point corresponds inversely to the $\log_{10}$-transformed Spearman's $p$ value. The false discovery rates (FDR) were calculated using SAS and significant values ($Q < 0.05$) are represented by a black border.

chronic infection prior to ART initiation (d35 p.i.) (Fig. 1i). Furthermore, at d58 post ATI, by which all RMs had experienced virologic rebound and PEG-IFNα therapy was no longer effective (Figs. 4c and 5a), the frequency of terminally differentiated NK cells in cytokine-treated RMs converged with levels observed in controls (Supplementary Fig. 10f); indicating that during viremic conditions prior cytokine therapy alone does not lead to persistent alterations in NK cell differentiation in pathogenic models of infection. Finally, the frequency and innate activity of terminally differentiated NK cells, both on-ART (d374 p.i.) and following ATI (d13), were positively correlated with the delay in rebound of plasma viremia, whereas the frequency of intermediate NK cells was associated with poor viral control following ATI (Fig. 5e). These analyses revealed a unique and strong relationship between the cytokine therapy-mediated differentiation of SIV-Env-specific, HLA-E-restricted NKG2a/c$^{low}$CD16$^+$ NK cells and their activity during ART with the subsequent delay in viral rebound following ATI.

## Discussion

Although historically underappreciated in curative approaches for chronic viral infections, these data demonstrate that HLA-E-restricted NK cell responses impact SIV control in vivo, as has been found in mouse models for other viral infections[33–35]. The formation of a terminally differentiated NK cell subset (NKG2a/c$^{low}$CD16$^+$) with robust innate and adaptive antiviral activities, which was found in nonpathogenic infections in AGMs[7], was blocked in pathogenic SIVmac infection in favor of intermediate differentiation (NKG2a/c$^{high}$CD16$^+$) with heightened pro-inflammatory potential[36]. Notably, we have demonstrated in a pathogenic model of infection that IL-21 and IFNα treatment during ART were effective in removing this block and in promoting NK cell terminal differentiation without altering their follicular homing or inducing a de novo expansion. As terminally differentiated NK cells were correlated with reductions in lymphoid replication competent virus during ART and the delay in viral rebound after ATI, these data support a role for SIV-Env-specific, HLA-E-restricted NK cells responses in controlling SIV in tissues. These data parallel earlier observations that robust NK cell responses in natural SIV hosts are a critical determinant of viral dissemination and control in secondary lymphoid tissue[37], which may represent a key factor mediating chronic inflammation and AIDS progression[38,39] in people living with HIV. It is plausible that viral control is influenced by a complex array or distinct parameters. For example, studies demonstrating CD8$^+$ T-cell-mediated viral control utilized an anti-CD8 depleting mAb (MT-807R1) that would also deplete NK cells as both express the CD8α chain[40–42]; however, this interplay could be resolved with depletions targeting the CD8β chain, which is only

expressed on T cells[43]. Furthermore, these models would underestimate the role of NK cells in viral control given our data demonstrate that NK cell terminal differentiation is blocked with pathogenic infection, which we postulate is a consequence of persistent inflammation and the cytokine environment in lymphoid tissue, or other factors influencing NK cell education, such as aberrant MHC expression and/or MHC peptide loading. As a caveat, although IL-21 therapy alone promoted NK cell maturation, without monotherapy control arms or further sampling we cannot determine if rIFNα treatment resulted in synergy or if NK cell responses were stably sustained between interventions, respectively; in addition, we cannot discern the role of PEG-IFNα during ATI in delaying viral rebound. Overall, these data demonstrate that AGM-like profiles of NK cell terminal differentiation, and also SIV-Env-specific, HLA-E-restricted activity, can be rescued via IL-21 and IFNα therapy in SIVmac-infected RMs, which impacts the size of the lymphoid replication competent virus during ART and viral recrudescence following ATI. As NK cell terminal differentiation was more strongly and consistently associated with viral control than the innate and HLA-E-restricted activity alone, these cells might exert additional antiviral properties, possibly including ADCC. As such, targeting NK cell differentiation states is a viable strategy when designing immunotherapy regimens to facilitate clinical HIV remission.

## Methods

**Study design.** Sixteen female Indian-origin RMs (*Macaca Mulatta*) were recruited to this study and housed at YNPRC (Supplementary Table 1). All animals were Mamu-B$^*$08$^-$ and -B$^*$17$^-$, whereas 4 RMs were Mamu-A$^*$01$^+$: RHr12, RNa12, RSp14, and RQb13. RMs were deemed pathogen-free and housed as previously describe[44]. RMs were i.v. infected with 300 TCID50 SIVmac$_{239}$ (Fig. 1a), which was purchased from Koen Van Rompay at UC-Davis. RMs were stratified into in vivo therapy cohorts balancing for their set point plasma viral loads at day 35 p.i. and their Mamu-A$^*$01 haplotype (Supplementary Table 1). At d35 p.i. RMs began a daily, s.c. triple formulation ART regimen consisting of tenofovir disoproxil fumarate (TDF; 5.1 mg/kg/d; Gilead Sciences), emtricitabine (FTC; 40 mg/kg/d; Gilead Sciences), and dolutegravir (DTG; 2.5 mg/kg/d; ViiV Healthcare)[18] that were obtained via a material transfer agreement (MTA). Nine RMs were administered two cycles of rhesus IL-21-IgFc (IL-21; 4 doses, once per week, s.c. 100 μg/kg) starting at day 42 p.i. and again at day 189 p.i. All animals with prior IL-21 therapy were subsequently administered rhesus IFNα-IgFc (rIFNα; once per week, s.c. 50,000 units) starting at day 323 p.i. (3X) and day 383 p.i. (2X; i.e., ART + IL-21 + IFNα, cytokine-treated). Cytokine-treated RM 172_10 was euthanized at day 66 p.i. due to rapid progression to AIDS-defining endpoints related to weight loss. Five RMs were utilized as ART-only controls (i.e., controls) and two RMs (RPk11 and RNa12) were projected to be controllers based on pre-established criteria[45]; hence they were excluded from analyses and not assigned to an experimental group but followed as a part of a study aimed at characterizing post-treatment control. Following ART ATI (day 402 p.i.), the eight remaining cytokine-treated RMs transitioned to human PEG-IFNα (7 doses, once every 6–8 days, s.c. 7 μg/kg) starting at day 5 post ATI. Animals were followed for 6 months following ATI and subjected to necropsy. The longitudinal characterization of ISGs via RNA-seq in PBMCs was performed in a historical cohort ($n = 6$;

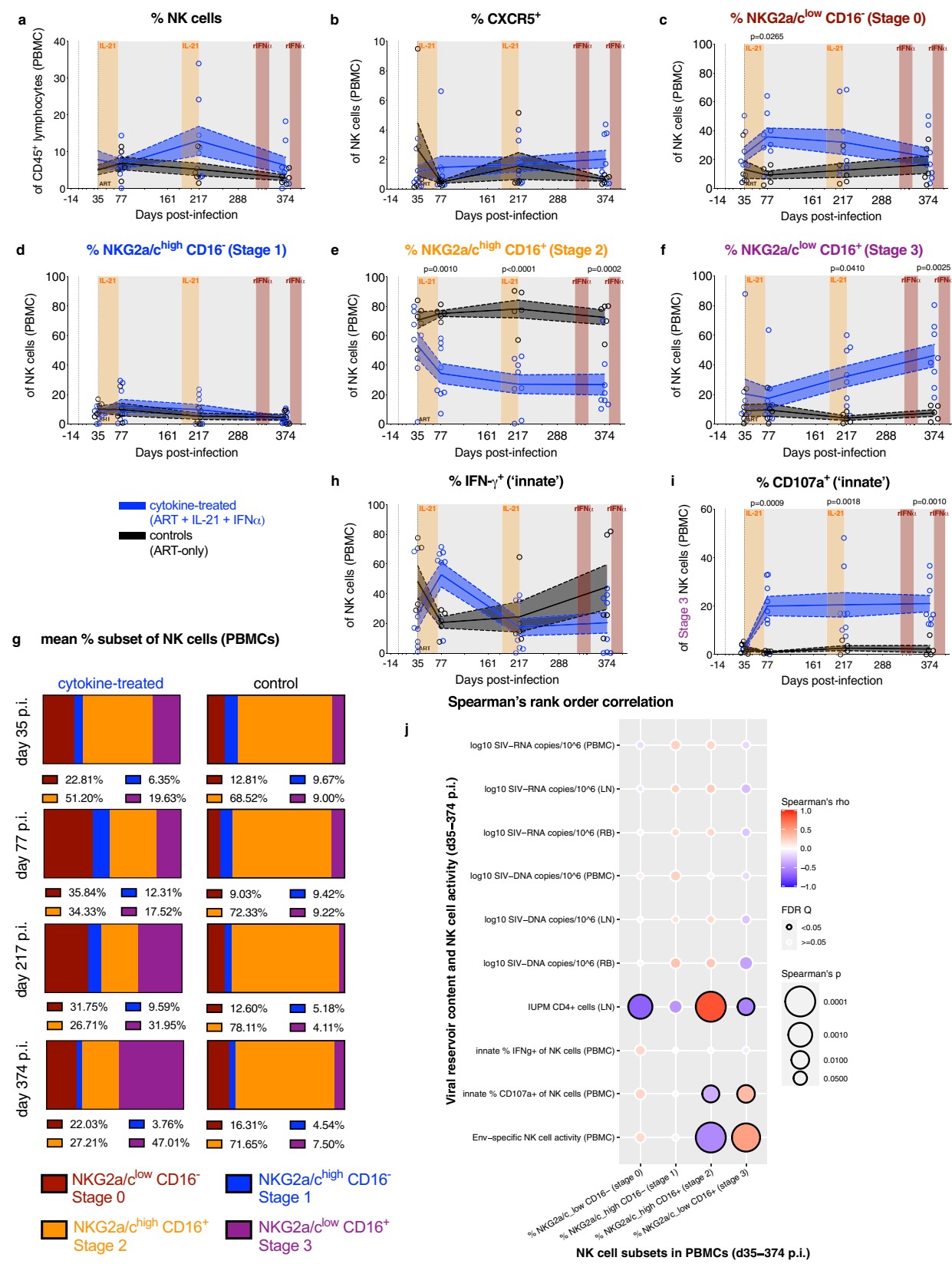

Supplementary Fig. 4)[23] in which RMs were infected i.v. with SIVmac[251] and treated with an ART regimen consisting of s.c. 3TC and TDF daily; intramuscular brecanavir weekly; and intramuscular cabotegravir once every 3 weeks.

**Study approval**. All animal experimentation was conducted following guidelines set forth by the Animal Welfare Act and by the NIH's Guide for the Care and Use of Laboratory Animals, 8th edition. All procedures were performed in accordance with institutional regulations and were approved by Emory University's Institutional Animal Care and Use Committee (permit 3000434). Animal care facilities are accredited by the US Department of Agriculture and the Association for Assessment and Accreditation of Laboratory Animal Care International. Proper steps were taken to minimize animal suffering and all procedures were conducted under anesthesia with follow-up pain management as needed.

**Fig. 3 Cytokine therapy promotes the maturation of NKG2a/c^low^CD16+ NK cells with enhanced ex vivo innate activity, which correlates with the content of lymphoid replication competent virus. a** The frequency of NK cells (CD45+CD20−CD3−NKG2a/c+) of PBMC CD45+ lymphocytes was longitudinally measured by flow cytometry, **b** as was their CXCR5 expression. The frequency of each differentiation stage of NK cells was determined based on the following definition: **c** Stage 0 (red, NKG2a/c^low^CD16−), **d** Stage 1 (blue, NKG2a/c^high^CD16−), **e** Stage 2 (orange, NKG2a/c^high^CD16+), and **f** Stage 3 (purple, NKG2a/c^low^CD16+). **g** The mean frequency of each NK cell differentiation stage from above was also re-visualized as a color-coded (as annotated below), parts-of-whole stacked bar plot for the cytokine-treated ($n = 8$; at left) and control RMs ($n = 5$; at right) over time (indicated at left). Flow cytometry was used to quantify the ex vivo innate frequency of **h** IFN-γ+ (intracellular) NK cells and **i** CD107a+ (surface) stage 3 (NKG2a/c^low^CD16+) NK cells. **a–i** Data from individual RMs (staggered open circles) are overlaid against the mean (solid line) ± SEM (shaded region within the dashed lines): control (ART-only, black; $n = 5$) and cytokine-treated (ART + IL-21 + IFNα, blue; $n = 8$). Treatment phases are indicated with the following background shading: IL-21 (orange), rIFNα (red), and ART (gray). Data were analyzed with two-sided (95% CI), two-way ANOVA with Bonferroni's correction with cross-sectional comparisons relative to controls. **j** The frequencies of each differentiation stage of NK cells in PBMCs (as indicated below) were correlated against levels of cell-associated and replication competent SIV content in tissue, and the ex vivo innate and Env-specific NK cell activities in PBMCs (as indicated at left) in all RMs ($n = 13$; days 35, 77, 217, and 374 p.i. as matched data are available). Per each correlation the two-tailed (95% CI) Spearman's rank correlation coefficient (rho) is represented as a double-gradient heatmap and the size of each data point corresponds inversely to the log₁₀-transformed Spearman's $p$ value. The false discovery rates (FDR) were calculated using SAS and significant values ($Q < 0.05$) are represented by a black border.

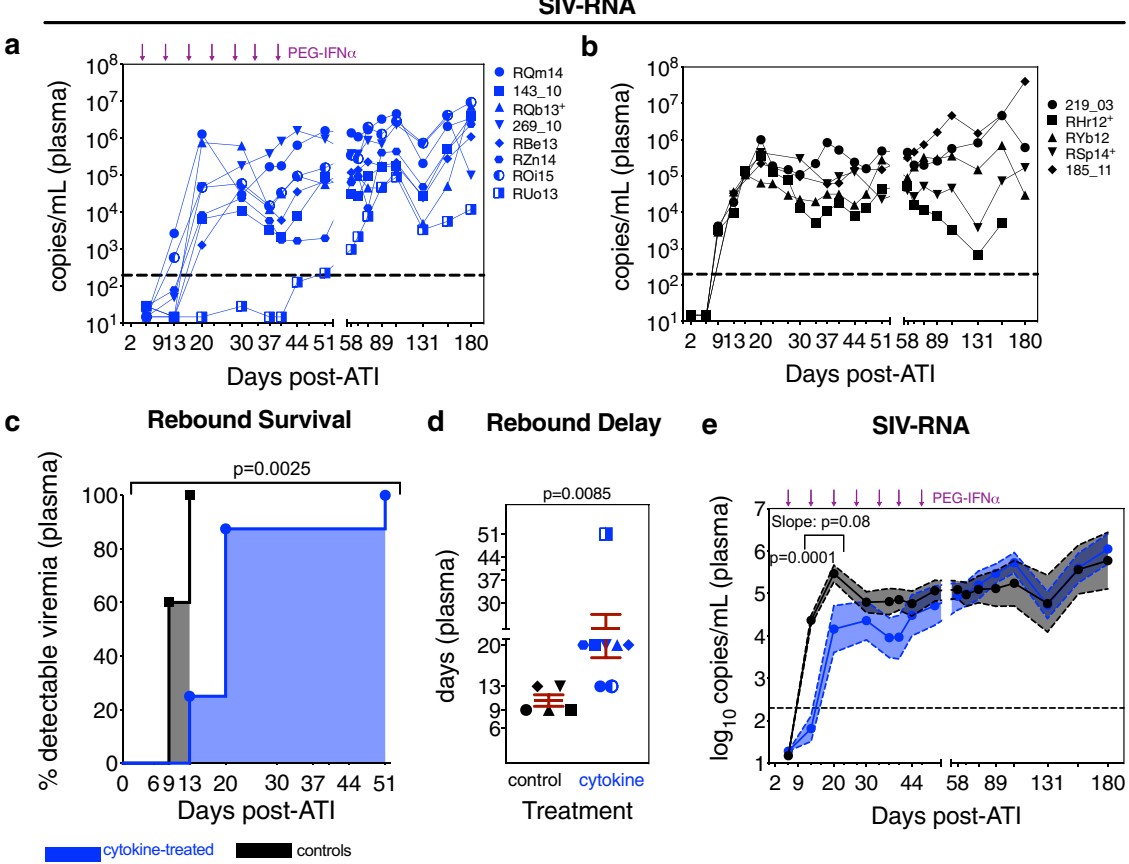

**Fig. 4 Cytokine therapy delays the rebound of plasma viremia following ATI.** Following ART analytical treatment interruption (ATI), the plasma SIV-RNA copies/mL were measured by qRT-PCR in each **a** cytokine-treated (PEG-IFNα with prior ART + IL-21 + rIFNα, blue; $n = 8$ RMs) and **b** control RMs (prior ART-only, black; $n = 5$ RMs). The horizontal dashed line (200 copies/mL) represents the threshold for virologic rebound and PEG-IFNα treatments are indicated by the purple arrows above. These kinetics of plasma viremia following ATI were then re-visualized as follows: **c** The delay in rebound of plasma viremia was represented as a treatment-stratified survival curve, which was analyzed with a Log-rank Mantel–Cox test. **d** The delay in viral rebound, in days, per each RM was represented as a color and shape-coded symbol overlaid against the mean ± SEM (red), which was analyzed with a two-sided (95% CI) Mann–Whitney $U$ test. **e** The log₁₀ SIV-RNA copies/mL are given as a longitudinal mean (solid line with closed circles) ± SEM (color-coded shaded region within the dashed lines), which was analyzed with a two-sided (95% CI), two-way ANOVA with Bonferroni's correction for multiple comparisons across treatments (d13 $p = 0.0001$), and a mixed-effects linear model was used to analyze the slope between d13 and d20 post ATI (as indicated by the bracket; $p = 0.08$).

**Tissue collection and processing**. Collections of peripheral blood (PB), RB punches, and LN biopsies were conducted longitudinally and upon necropsy (Fig. 1a) as previously described[44]. EDTA PB was used for complete blood counts, and plasma was separated by centrifugation within 1 h of phlebotomy. PBMCs were isolated from PB by density gradient centrifugation (Ficoll-Paque Premium, GE Healthcare). RB punches were obtained by inserting an anoscope a short

distance into the rectum and 20 punches were collected using a biopsy forcep. To obtain gut-derived lymphocytes, RB punches were digested with 1 mg/mL collagenase for 2 h at 37 °C with agitation, and then filtered with a 100-μm strainer to remove residual tissue fragments. For LN biopsies, the skin over the axillary or inguinal region was clipped and surgically prepped. An incision was then made in the skin and the LN was exposed by blunt dissection and excised over clamps. LNs

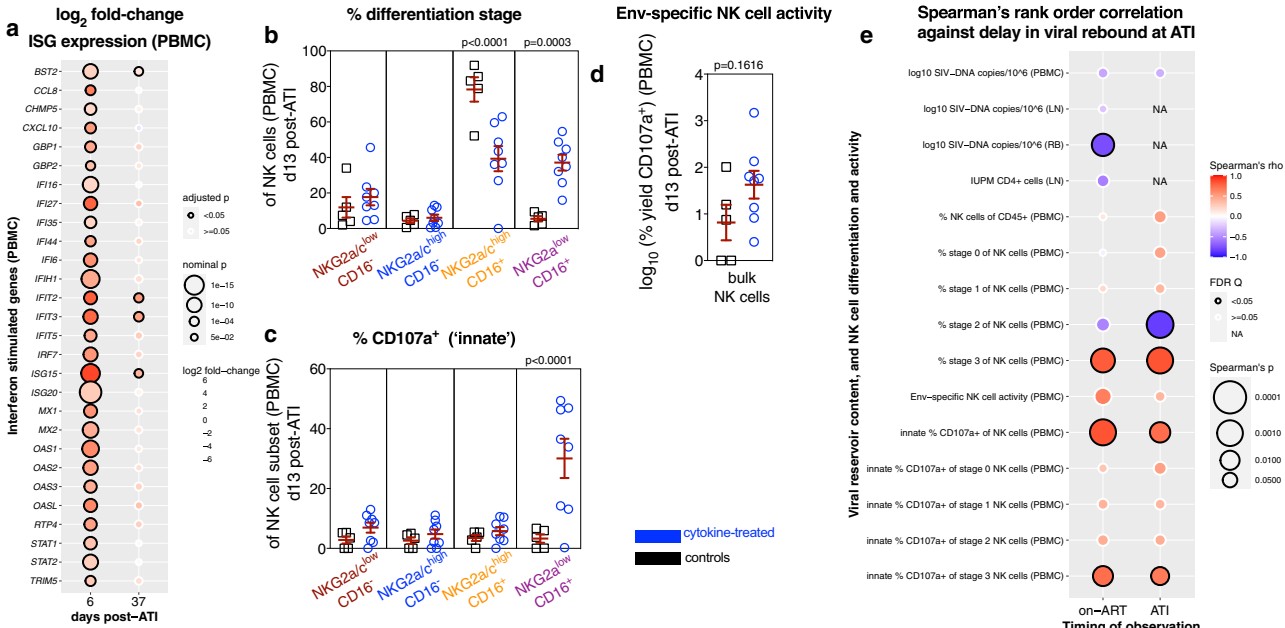

**Fig. 5 The formation and activity of NKG2a^lowCD16^+ NK cells correlate with viral recrudescence following ATI. a** From PBMCs taken 24 h following the first (d6 post ATI) and the fifth PEG-IFNα dose (d37 post ATI), the expression of interferon-stimulated genes was calculated as a cross-sectional log$_2$ fold-change between cytokine-treated ($n = 8$) and control ($n = 5$) RMs, which is represented as a double-gradient heatmap. The size of each data point corresponds inversely to the log$_{10}$-transformed nominal *p* value with significant ($p < 0.05$) adjusted *p* values indicated by a black border. Using DESeq2, data were analyzed with a two-sided (95% CI) Wald test using the Benjamini–Hochberg method for multiple comparisons. **b** In PBMCs at d13 post ATI, the distribution of the differentiation subsets was measured by flow cytometry as a frequency of NK cells: Stage 0 (red, NKG2a/c^lowCD16^−), Stage 1 (blue, NKG2a/c^highCD16^−), Stage 2 (orange, NKG2a/c^highCD16^+), and Stage 3 (purple, NKG2a/c^lowCD16^+). NK cells isolated from PBMCs at d13 post ATI were utilized to determine **c** the frequency of ex vivo innate activity (CD107a surface expression) or **d** the Env-specific activity upon co-culture with K562 cells expressing HLA-E loaded with SIVmac Env peptides. **b–d** Data from individual cytokine-treated (PEG-IFNα with prior ART + IL-21 + rIFNα, blue; $n = 8$) and control (prior ART-only, black; $n = 5$) RMs (open symbols) are overlaid against the mean ± SEM (in red) and were analyzed with **b**, **c** two-sided (95% CI), two-way ANOVA with Bonferroni's correction for cross-sectional comparisons relative to controls or **d** with a two-sided (95% CI) Mann–Whitney *U* test. **e** The delay in the rebound of plasma viremia was correlated against measures of SIV content, NK cell differentiation, and NK cell activity (as indicated at left; $n = 13$) from the final on-ART measurement (d339–374 p.i.) or during rebound following ATI (d6–13 post ATI). Per each correlation the two-tailed (95% CI) Spearman's rank correlation coefficient (rho) is represented as a double-gradient heatmap and the size of each data point corresponds inversely to the log$_{10}$-transformed Spearman's *p* value. The false discovery rates (FDR) were calculated using SAS and significant values ($Q < 0.05$) are represented by a black border. Correlations for which data did not exist during that experimental phase are indicated as "NA".

were segmented using a sterile scalpel; macerated over RPMI 1640 medium supplemented with 10% heat-inactivated fetal bovine serum (FBS; Gemini Bio), 100 U/mL penicillin, and 100 μg/mL streptomycin; and filtered through a 100-μm strainer to isolate mononuclear cells. Tissue segments of LN and 2–4 RB punches were flash frozen in dry ice for SIV-DNA analysis, whereas processed mononuclear cells were cryo-preserved in 10% dimethyl sulfoxide (DMSO) in FBS.

**Flow cytometric analysis**. Fourteen-parameter flow cytometric analysis was performed on fresh PBMCs and mononuclear cells derived from LN biopsies and RB punches. Samples were stained utilizing standard procedures employing clones of anti-human mAbs that we have shown to be cross-reactive in RMs[20,44,46,47] and are validated in databases maintained by NHP Reagent Resource (MassBiologics: https://www.nhpreagents.org/ReactivityDatabase). The following antibodies were utilized for the longitudinal staining panel per T-cell and B-cell characterization: anti-CD21-PE (clone B-ly4, 10 μL, cat. 555422), anti-CD28-PE-CF594 (clone CD28.2, 5 μL, cat. 562296), anti-CD95-PE-Cy5 (clone DX2, 10 μL, cat. 559773), anti-CCR7-PE-Cy7 (clone 3D12, 7.5 μL, cat. 557648), anti-CD45RA-APC (clone 5H9, 10 μL, cat. 561210), anti-Ki-67-AL700 (clone B56, 5 μL, cat. 561277), and anti-CD3-APC-Cy7 (clone SP34-2, 5 μL, cat. 557757) all from BD Biosciences; anti-HLA-DR-BV570 (clone L243, 5 μL, cat. 307638), anti-CD27-BV605 (clone O323, 10 μL, cat. 302830), anti-CD20-PerCP-Cy5.5 (clone 2H7, 5 μL, cat. 302326), and anti-CD4-BV421 (clone OKT4, 2.5 μL, cat. 317438) all from BioLegend; anti-CD38-FITC (clone AT-1, 5 μL, cat. 60131FI) from STEMCELL Technologies; and anti-CD8-Qdot705 (clone 3B5, 1 μL, cat. Q10059) and Live/Dead Fixable Aqua (AmCyan, 2 μL of 1:20 PBS dilution, cat. L34957) from Thermo Fisher Scientific (Supplementary Fig. 1a). A secondary panel using a similar parental gating strategy, with the exception of using anti-CD4-BV605 (clone OKT4, 5 μL, cat. 317438 from BioLegend), was utilized to measure markers of cytotoxicity, such as anti-T-bet-PE (clone eBio4B10, 5 μL, cat. 12-5825-82 from eBioscience; Supplementary Fig. 1d). Stain volumes are given per 10$^6$ live mononuclear cells or per 4–5 processed RB punches. Cells were fixed and permeabilized

with a Cytofix/Cytoperm kit (BD Biosciences). Flow cytometry acquisition was performed on a minimum of 120,000 live CD3$^+$ T cells for PBMC and LN, and a minimum of 50,000 live CD3$^+$ T cells in RB. Data acquisition was performed on an LSR II (BD Biosciences) using FACS DiVa software (v8.0.1) and analyzed using FlowJo software (version 9.9.6; TreeStar). Gates for activation (HLA-DR$^+$CD38$^+$) and proliferation in memory CD4$^+$ T cells and memory CD8$^+$ T cells were set based on their expression within the corresponding naive subset.

Cryo-preserved PBMCs were stained as previously described[37] to characterize NK cell subsets using a panel of the following mAbs: anti-CD3-V500 (clone SP34-2, 5 μL, cat. 560770), anti-CD45-PerCP (clone D058-1283, 5 μL, cat. 558411), and anti-CD69-APC-Cy7 (clone FN50, 5 μL, cat 557756) all from BD Biosciences; anti-CD8-VioBlue (clone BW135/80, 2.5 μL, cat. 130-094-152), anti-NKp80-FITC (clone 4A4.D10, 8 μL, cat. 130-094-843), anti-NKp30-APC (clone AF29-4D12, 8 μL, cat. 130-092-484) all from Miltenyi Biotec; and anti-CD20-AL700 (clone 2H7, 8 μL, cat. 56-0209) from eBioscience; anti-TIM-3-PE-Cy7 (clone F38-2E2, 5 μL, cat. 345014); and anti-NKG2a-PE (clone Z199, 8 μL, cat. IM3291U), anti-CD16-ECD (clone 3G8, 5 μL, cat. 6607111), and anti-NKp46-PE-Cy5 (clone BAB281, 8 μL, cat. A66902) all from Beckman Coulter (Supplementary Fig. 8). A secondary immunophenotyping panel consisted of the following mAbs: anti-CD3-V500 (clone SP34-2, 5 μL, cat. 560770), anti-CD45-PerCP (clone D058-1283, 5 μL, cat. 558411), anti-CD95-APC (clone DX2, 5 μL, cat. 558814), anti-CD107a-PE-Cy5 (clone HA43, 5 μL, cat. 555802), and anti-CCR7-BV711 (clone 3D12, 5 μL, cat. 563712) all from BD Biosciences; anti-CD8-VioBlue (clone BW135/80, 2.5 μL, cat. 130-094-152) from Miltenyi Biotec; anti-CD16-ECD (clone 3G8, 5 μL, cat. 41116015) and anti-NKG2a-PE (clone Z199, 8 μL, cat. IM3291U) from Beckman Coulter; anti-HLA-DR-PE-Cy7 (clone L243, 5 μL, cat. 307616) and anti-PD-1-APC-Cy7 (clone EH12.2H7, 5 μL, cat. 329922) from BioLegend; and anti-CXCR5-FITC (clone MU5UBEE, 5 μL, cat. 15566616) from eBioscience. A variant of this panel used anti-HLA-E-PE (clone 3D12HLA-E, 6 μL, cat. NBP2-00277) from R&D Systems. Representative gating strategies are given in Supplementary Fig. 1b, c and representative CD107a stains are given in Supplementary Fig. 6d–f. The anti-NKG2a mAb recognizes both NKG2a and NKG2c in macaques. Stain volumes are given per 10$^6$ live

mononuclear cells upon thawing. Cells were fixed and permeabilized with a Cytofix/Cytoperm kit (BD Biosciences) and intracellular stain was incubated a 4 °C for 15 min and anti-IFN-γ-AL700 (clone GB11, 8 μL, cat. 560213). For the NK cell panels, flow cytometry acquisitions were done on a LSR II (BD Biosciences) and were analyzed using FlowJo software (version 10.4.2; TreeStar).

**Intracellular cytokine staining**. Th17 and Th22 cells were defined as the frequency IL-17 and IL-22 producing CD4+ T cells upon ex vivo stimulation with PMA and ionomycin[20,22]. Fresh mononuclear cells derived from RB punches were incubated for 4 h at 37 °C in RPMI 1640 medium supplemented with 10% FBS containing PMA (80 ng/mL; cat. P8139, Millipore Sigma), calcium ionophore A23187 (500 ng/mL; cat. C9275, Millipore Sigma), brefeldin A (10 μg/mL; cat. 420601, BioLegend), and GolgiStop with monensin (7 × 10$^{-4}$ dilution; cat. 554724, BD Biosciences). Samples were stained utilizing standard procedures employing clones of anti-human mAbs that were cross-reactive in RMs as validated by NHP Reagent Resource (MassBiologics): anti-CD8-PE-CF594 (clone RPA-T8, clone 562282), anti-CD95-PE-Cy5 (clone DX2, 10 μL, cat. 559773), anti-IFNγ-PE-Cy7 (clone B27, 5 μL, cat. 557643), anti-TNFα-AL700 (clone Mab11, 1 μL, cat. 557996), and anti-CD3-Cy7APC (clone SP34-2, 5 μL, cat. 557757) all from BD Biosciences; anti-CD4-BV421 (clone OKT4, 4 μL, cat. 317434) and anti-IL-2-BV605 (clone MQ1-17H12, 1 μL, cat. 500332) both from BioLegend; and anti-IL-17-FITC (clone eBio64DEC17, 5 μL, cat. 53-7179-42), anti-IL-22-APC (clone IL22JOP, 5 μL, cat. 17-7222-82), and Live/Dead Fixable Aqua (AmCyan, 2 μL at 1:20 PBS dilution, cat. L34957) from Thermo Fisher Scientific (Supplementary Fig. 3a). Stain volumes are given per test (approximately 5–10 RB punches). Cells were fixed and permeabilized with a Cytofix/Cytoperm kit (BD Biosciences). The intracellular stain was incubated for 1 h at room temperature in the dark. Data acquisition was performed on a minimum of 50,000 live CD3+ T cells on an LSR II (BD Biosciences) using FACS DiVa software (v8.0.1) and analyzed using FlowJo software (version 9.9.6; TreeStar).

**Determination of plasma viremia**. Six-replicate reaction hybrid qRT-PCR assay was performed to determine SIV-RNA copies per mL of EDTA plasma (i.e., viral loads) as previously described[48] with a limit of detection of 30 copies/mL. While primers target a highly conserved viral sequence in SIVmac$_{239}$ *gag* (Supplementary Table 2), unconventional neutral bases are incorporated to avoid biasing in measuring off-target viral sequences. A Poisson method was employed to calculate values for which not all reactions were positive, which was otherwise calculated by interpolating individual cycle threshold values on a standard curve. Primers and probe sequences are given in Supplementary Table 2.

**Quantification of cell-associated SIV-DNA and -RNA**. Analyses were performed on cryo-preserved PBMCs, snap frozen LN biopsies, and snap frozen RB punches. Cell-associated SIV-DNA and -RNA were extracted with an AllPrep DNA/RNA Mini Kit (Qiagen), and were quantified via hybrid real-time/digital qPCR and RT-qPCR assays for SIV-DNA and -RNA, respectively, in ten replicates with single-copy clinical sensitivity as previously described[49]. Viral copy numbers were normalized based on genomic CCR5. A Poisson method was employed to calculate samples for which there were fewer than ten positive replicates for amplification, which was otherwise calculated by interpolating a standard curve. Primers and probe sequences are given in Supplementary Table 2.

**Quantitative viral outgrowth assay (QVOA)**. Restricted by cellular yields, the QVOA was limited to four controls (ART-only) and six cytokine-treated RMs (ART + IL-21 + IFNα). CD4+ cells were positively selected utilizing a MACS CD4 MicroBeads kit for NHPs (cat. 130-091-102, Miltenyi Biotec) from LN cryo-preserved mononuclear cells. After a 1 h rest in RPMI 1640 medium supplemented with 10% FBS, cells were stimulated for 16 h at 37 °C with IL-2 (recombinant human, 20 ng/mL; Tonbo Biosciences), anti-CD28 mAb (clone CD28., 5 μg/mL; BD Pharmingen), and anti-CD2 mAb (clone RPA-2.10, 5 μg/mL; BioLegend) in a 96-well polystyrene plate coated with anti-CD3 mAb (clone SP34-2, 1 μg per well; BD Biosciences). Stimulated CD4+ cells were washed twice and co-cultured in a 1:1 ratio with 174xCEM cell line (HIV Reagent Program managed by ATCC)[50] in three serial dilutions in triplicates ranging from 1 × 10$^6$ to 0.1 × 10$^6$ cells per well. The cells were cultured at 37 °C under 5% CO$_2$ in complete RPMI 1640 with 4 mM L-glutamine supplemented with 10% heat-inactivated FBS (Gemini Bio-products), penicillin (50 U/mL), streptomycin (50 μg/mL), and IL-2 (20 ng/mL). Cultured cells were fed weekly with fresh medium supplemented with IL-2 (20 ng/mL final concentration), and harvested and analyzed at day 10 and 25. Positive wells were identified by the expression of intracellular SIV-Gag p27 via flow cytometry[51] and the amplification (>10×) of SIV-RNA (copies/mL) in the supernatant as determined by 6-replicate reaction qRT-PCR between day 10 and 25. The frequency of cells producing replication competent virus was determined by the maximum-likelihood method[52] utilizing IUPMStats[53] and was expressed in infectious units per million.

**Quantification of IFN-γ response upon SIV-Gag and -Env stimulation**. ELISpots were performed as previously described[46]. Briefly, cryo-preserved PBMCs were thawed, rested, and triplicate plated (4 × 10$^5$ cells per well) for an enzyme-linked immune absorbent spot (ELISpot) assay. Cells were either mock stimulated (DMSO); stimulated with overlapping SIVmac$_{239}$ Gag or Env peptides (HIV

Reagent Program managed by ATCC; 1 μg/mL); or stimulated with Concavalin A (Sigma; 2.5 μg/mL). Plates were incubated for 24 h at 37 °C and the number of IFN-γ secreting cells was quantified per the manufacturer's instructions using the Monkey IFN-γ ELISpot$^{PLUS}$ kit (MABTECH). Spot forming units (SFU) per well were counted on an ELISpot plate reader and represented as SFU per million cells. The IFN-γ response given per stimulation condition is an average of triplicate reactions with the average DMSO background subtracted.

**Ex vivo innate NK cell activity assay**. NK cell degranulation activity was determined through expression of cell surface CD107a, as previously described[54]. Cryo-preserved PBMCs were labeled with anti-NKG2a/c-PE (clone Z199, cat. IM3291U; Beckman Coulter) conjugated with anti-PE MicroBeads (Miltenyi Biotec) and magnetically isolated according to the manufacturer's instructions. NKG2a/c+ cells were incubated overnight at 37 °C with 5% CO$_2$ in RPMI supplemented with 10% (v/v) FBS, 100 U/mL IL-2, and 10 ng/mL of IL-15, and then analyzed for cell surface CD107a expression via flow cytometry.

**Env-specific NK cell activity assay**. K562 cells devoid of MHC-I stably (ATCC) expressing HLA-E$^*$0101 (K562-E$^*$0101; Applied Biological Materials Inc.) cells were incubated with 50 μM of SIVmac$_{239/251}$ Env peptide (NQLLIAILL) at 26 °C for 15–20 h[7]. Isolated NK cells were cultured for 6 h in the presence of an anti-CD107a-PE-Cy5 mAb (clone HA43, 5 μL, cat. 555802; BD Biosciences), either alone (NK), or co-cultured at a 5:1 ratio with 2 × 10$^4$ K562-E$^*$0101 cells that were either unpulsed (NK+K562$^*$E) or loaded with SIVmac$_{239/251}$ Env peptide (NK+K563$^*$E+ENV). GolgiStop and GolgiPlug (BD Biosciences) was added 1 h following culture initiation. The frequency of NK cells expressing surface CD107a was measured by flow cytometry per each culture condition to assess the background (%CD107a$^+_{NK}$), the maximum (%CD107a$^+_{NK+K562^*E}$), and the peptide-specific degranulation activity (%CD107a$^+_{NK+K562^*E+ENV}$; Supplementary Fig. 6). From these measurements the Env-specific NK cell activity was calculated as previously described[30,55]:

$$\log_{10}\left[100 \times \frac{\%CD107a^+_{NK+K562^*E+ENV} - \%CD107a^+_{NK}}{\%CD107a^+_{NK+K562^*E} - \%CD107a^+_{NK}}\right] = \text{Env} - \text{specific NK cell activity} \quad (1)$$

**Production and testing of rhesus rIL-21-IgFc**. A fusion protein of rhesus IL-21-IgFc (IL-21) fusion protein was generated as previously described by the Resource of Nonhuman Primate Immune Reagents at New Iberia Research Center and were provided via MTA[20,22,56,57]. Using the Drosophila S2 system, a fusion protein was produced between rMamuIL-21 and a macaque IgG2 Fc, which was mutated (L235A and P331S) to block complement or Fc receptor binding[58,59]. IL-21 was isolated to >95% purity by Protein-G sepharose affinity chromatography, dialyzed against PBS, and tested for sterility and confirmed endotoxin free.

**Production and testing of rhesus IFNα-IgFc**. Using the Drosophila S2 system, a fusion protein was produced, as previously described[56,57], between a recombinant RM IFNα2 and a macaque IgG2 Fc[19] by the Resource of Nonhuman Primate Immune Reagents at New Iberia Research Center and was provided via MTA. The Fc was mutated at two positions (L235A and P331S) to block complement or Fc receptor binding[58,59]. IFNα-IgFc was isolated to >95% purity by Protein-G sepharose affinity chromatography, dialyzed against PBS, lyophilized, and tested for sterility and confirmed endotoxin free. Activity was verified as 264 × 10$^3$ U/mg as determined by Vero/EMCV bioassay.

**Peginterferon alfa-2A**. Pharmaceutical-grade human PEG-IFNα was purchased at cost (PEGASYS®, Roche), which was previously shown to be well tolerated in SIV-infected macaques[21,60].

**RNA sequencing**. RNA was extracted from PBMCs stored at −80 °C in RLT buffer with 1% 2-mercaptoethanol using RNeasy Mini kits (QIAGEN, CA) with DNase digest and QIAcube automation stations. Extracted RNA was quantified using a NanoDrop 2000 spectrophotometer (Thermo Scientific Inc. Wilmington, DE) and the quality was assessed by Bioanalyzer analysis (Agilent Technologies, Santa Clara, CA). Ten nanogram of total RNA was used as input for cDNA amplification using 5′ template-switch PCR with the Clontech SMART-Seq v4 Ultra Low Input RNA kit. Amplified cDNA was fragmented and appended with dual indexed bar codes using Illumina NexteraXT DNA Library Prep kits. The amplified libraries from both sets were validated by capillary electrophoresis on the Agilent 4200 TapeStation. The libraries were normalized, pooled, and sequenced on the Illumina HiSeq 3000 system employing a single-read 101 cycles run at average read depths of 30 × 10$^6$ reads per sample. Reads were mapped to the MacaM version 7 assembly of the Indian rhesus macaque genomic reference[61] with RhesusGenome using STAR (version 2.5.2b) with default alignment parameters[62] (https://www.unmc.edu/rhesusgenechip/index.htm). Abundance estimation of raw read counts per transcript was done internally with STAR using the HTSeq-count algorithm[63].

**Statistics and reproducibility**. Statistical tests were two-sided and *p* values ≤0.05 (95% confidence interval, CI) were considered statistically significant for each of the

specific statistical comparisons. All experiments were performed as a single technical replicate unless otherwise noted in the Methods (i.e., qRT-PCR, IFN-γ ELISpot, and QVOA). No assays were repeated as independent experiments. Data were tested for Gaussian distribution using the D'Agostino–Pearson omnibus normality test. Data showing continuous outcomes are represented as mean ± SEM. Two-way ANOVAs and/or mixed-effects models, in the event of absent data points, were performed with Bonferroni's correction for multiple comparisons. Correlations were performed two-sided with a non-parametric Spearman correlation and were fitted with a simple linear regression. Comparisons of survival curves were conducted with a Log-rank (Mantel–Cox) test. All of the above analyses were conducted using GraphPad Prism version 8.1.2. Using SAS, Spearman's $p$ values were adjusted for multiple comparisons using the stepdown Bonferroni[64], Hochberg[65], and false discovery rate (FDR)[66] methods. Correlation and RNA-seq data were visualized using ggplot2 (version 3.3.2) in RStudio (version 1.4.1103) with custom code. The distribution of cytokine co-expression (i.e., Boolean logical gates of IL-17 and IL-22 expression within CD4+ T-cells) was analyzed with a Permutation test ($10^3$ iterations) in SPICE version 6[67]. Rates of increase in $\log_{10}$ SIV-RNA copies per mL of plasma post ATI were obtained using a mixed-effects linear model specifying that data follow a linear regression over time, with a random intercept for each animal. The mean slope and mean linear increase were estimated and compared between treatment conditions within the framework of the mixed-effects linear model[68]. DESeq2 version 1.22.1R package[69] was used to produce normalized read counts and compute the differential expression estimation using the Wald test. Multiple-test correction was performed with the Benjamini–Hochberg method and a FDR <0.05 was used to indicate statistical significance.

**Reporting summary.** Further information on research design is available in the Nature Research Reporting Summary linked to this article.

## Data availability

Materials provided via MTA were supplied without restrictions on use. Source data are provided for the performed correlations, as are the statistical readouts for the correlations and RNA-seq analyses. RNA-seq data related to Figs. 1f and 5a, and Supplementary Fig. 4 are publicly available in GenBank (https://www.ncbi.nlm.nih.gov/geo/query/acc.cgi) under GEO accession GSE163443 with subseries GSE163440, GSE163441, and GSE163442. The MacaM version 7 assembly of the Indian rhesus macaque genomic reference is publicly available at https://www.unmc.edu/rhesusgenechip/index.htm. Source Data are provided with this paper.

## Code availability

Other data that support the findings of this work, including custom ggplot2 (version 3.3.2) code for data visualization (RStudio version 1.4.1103) of Figs. 1f, 2h, 3j, 5a, e, and Supplementary Fig. 4, are available from the corresponding author on reasonable request. Source Data are provided with this paper.

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

## Acknowledgements

We thank Stephanie Ehnert and Christopher Souder from Research Resources, and Sherrie Jean and Jennifer Wood from Veterinary Medicine at Yerkes National Primate Research Center (YNPRC) for providing animal and veterinary care, respectively. We gratefully acknowledge Katie Kitrinos at ViiV Healthcare for supplying DTG, and Romas Geleziunas at Gilead Sciences for supplying FTC and TDF. Both the rhesus rIL-21-IgFc and rhesus IFNα-IgFc were kindly supplied by Kenneth Rogers at the Resource for Nonhuman Primate Immune Reagents. We would also like to thank Mathias Lichterfeld, Pilar Garcia Broncano, and Xiaodong Lian of the Ragon Institute, and William Bosche at Leidos Biomedical Research for technical assistance. The SIVmac239 strain used to infect the RMs was kindly provided by Koen Van Rompay of UC-Davis, and the 174xCEM cell line was supplied by Dr. Peter Cresswell via the HIV Reagent Program. We thank Richard Dunham at ViiV Healthcare for providing access to historical RNA-seq data from longitudinal SIV infection. This work was supported by the NIAID, NIH under award number R01AI116379 to M. P. and R01AI143457 to M. M.-T. Support for this work was also provided by ANRS and the Fondation J. Beytout to M. M.-T.; NCRR, NIH award 5R24RR016988 to F. V. and the Resource for Nonhuman Primate Immune Reagents; ORIP/OD award P51OD011132 to YNPRC; and NCI, NIH award HHSN261200800001E and 75N91019D00024 to Leidos Biomedical Research. P.R. was recipient of a PhD fellowship from the University Paris Diderot, Sorbonne Paris Cité. The content of this publication does not necessarily reflect the views or policies of the Department of Health and Human Services, nor does mention of trade names, commercial products, or organizations imply endorsement by the US Government.

## Author contributions

J.H. contributed to conceptualization, methodology, formal analysis, investigation, writing (original draft, review, and editing), and visualization. N.H. performed the phenotyping and ex vivo functional assays on NK cells and contributed to conceptualization, methodology, investigation, data analyses, and writing (review and editing). L.M. contributed to conceptualization, methodology, and investigation. G.T., A.A. U., and S.B. performed RNA-seq analyses. C.K. and H.W. performed longitudinal processing and flow cytometry of tissues. P. R. contributed to methodology and investigation. C.G. performed the ELI-Spot experiments. N.S. and K.E. performed statistical analyses of the correlations and viral rebound kinetics. F.V. provided investigational compounds. J.L. measured virus content in plasma and tissue. G.S. contributed to conceptualization. B.J. contributed to methodology. M.M.-T. contributed to conceptualization, methodology, writing (review and editing), supervision, and funding acquisition. M.P. contributed to conceptualization, methodology, resources, writing (original draft, review, and editing), supervision, and funding acquisition.

## Competing interests

The authors declare no competing interests.
