## [Peer Review File · Nature Communications]

REVIEWER COMMENTS

Reviewer #1 (Remarks to the Author):

Harper and Huot et al utilize a pathogenic model of SIV infection to determine the impact of IL-21 and IFN α cytokine therapy during ART on T cell activation, viral reservoirs and NK subsets, and following ATI. They identify changes in NK cell maturation states and cytolytic activity associated with cytokine therapy, which may be associated with viral reservoir size/rebound during ATI. The paper is well written, although the findings are occasionally overstated, and the figures are clear and well presented.

Major Comments:

1. There are a few occasions where the findings are overstated and are not completely supported by the data. It would be helpful to rephrase. For example:
 - a. In the abstract the authors state that the "cytolytic activity of NKG2a/c low CD16+ NK cells... predicted viral rebound delay following ATI". While the authors demonstrate a correlation between NK CD107a activity and days to rebound, this observational study is not powered for prediction.
 - b. Line 130-131 – "Altogether this implies that IL-21 regulates SIV persistence in the LN principally through an NK-mediated mechanism". As above, correlations are insufficient to support the strength of this claim.
 - c. Line 197 - "this shows that NK cells have the capacity to mediate SIV control in tissues". As above.
2. One of the key interesting findings for is the cytotoxic activity of NK cell subsets with cytokine therapy. However, as currently presented this data is challenging to interpret.
 - a. Could the authors explain why the data for this assay shown as log₁₀ transformed? From the M&M (without the companion paper accessible) the final readout appears to be flow cytometric detection of CD107a.
 - b. If this is indeed a flow-based assay, example plots are needed for interpretation.
 - c. It would also be useful to add what the control for this assay is (e.g. unpulsed cell line?).
 - d. Figure 4G – there is a solid grey box at CD107a+ NK vs d13 ATI, suggesting that the data does not exist for total NK. However, data is available for the NK subsets. Could the authors clarify why this data is missing please?
3. Much of Figure 4 is built around a delayed rebound during ATI following cytokine therapy. However, as the authors note, this has been seen when pegIFN α alone was administered. This does not invalidate the potential role for NKs and subsets, but this section should be rephrased to highlight the pegIFN α caveat upfront (rather than lines 181-183), and to focus on the NK findings.
4. In Figures 3C-F and 3G the authors nicely demonstrate the changes in NK subsets following cytokine therapy. However, the current analysis does not assess a change in frequency of these NK subsets, but a change in proportion with NK cells.
 - a. It would be helpful to clarify this in the text (line 141).
 - b. Modifying this analysis to ask what the percentage of these NK subsets is from CD45+ cells (rather than NK) may provide further insight to this finding. How do the proportional changes relate to the increase in total NKs? For example, when assessing a "block" between Stages 3 and 2, is the frequency of Stage 3 NK from CD45+ expanded, while Stage 2 is similar? Or is Stage 2 still lost? This could alter how this data is interpreted.
 - c. Given the importance of the definition of these NK subsets, raw plots to see the gating strategy are needed here.
 - d. Figure 3I – the increase in CD107a+ Stage 3 NK cells is really striking. However, the authors have not shown the data to support their claim (Line 158-160) that this population is solely responsible for CD107a+ expression by NKs. This would be shown by gating first on the CD107a+ NK cells, and then performing the subset analysis.
5. For the heatmaps (e.g. Figure 2H) all of the data (including the rho for non-significant p values) should be shown. Recent examples of alternative visualization approaches include Mathew et al Science 2020 and Moderbacher et al Cell 2020. Are these nominal p-values or has this been corrected for multiple comparisons? This may impact interpretation of this data.

Minor comments:

6. How were the gates for Figure 1E HLA-DR/CD38 determined? Is the same result seen when more stringent gates are applied (for example based on the naïve population)?
7. In Fig 1E, the control animal (RSp14) has few CD4 T cells at the initiation of ART in comparison to the cytokine-treated animal. Was the CD4 nadir equivalent between treatment arms?
8. The language in lines Line 118-120 around Figure 2G is a little confusing when compared to Figure 1A, the M&M and the figure legend. Could the authors clarify what treatment was received by the controls in 2G: are these ART-only, or did they receive rIFN α ?
9. How sustained is the change in NK activity and subset distribution? Is it still present when all animals rebound at Day ~51 post-ATI? If samples are not be available or experiments are not possible, some discussion in the text would be interesting here.

Reviewer #2 (Remarks to the Author):

In their manuscript "IL-21 and IFN α therapy rescues the maturation of terminally-differentiated NK cells and limit the SIV reservoir in ART-treated macaques", Harper et al. follow the effects of sequential rIL-21 and rIFN α treatment on the maturation of NK cells in the setting of pathogenic SIVmac239 infection. This is of particular interest because modulation of NK cell function is often overlooked when considering therapies to drive HIV-remission. The premise of the work is based on sound science and a wealth of previous knowledge gained from the study of NK cells during SIV/ART in rhesus macaques. The significance of the work is also high, since altering NK phenotypes in SIV-infected rhesus macaques to mimic that of non-pathogenic infected NHP (i.e. AGM) could lead to significant reductions in HIV morbidity and mortality. The results section is the only problematic section of the paper, and requires revisions and additional data to convince reviewer/author as to the veracity of the data.

Weaknesses:

- 1) Rectal biopsy needs to be appropriately abbreviated in the main text (line 91), and not just in the methods.
- 2) How were PBMC time points chosen for different assays? For instance, figure 1d shows many data points for flow staining of PBMC, but then figure 1h shows a total of four timepoints of PBMC staining per cohort?
- 3) Viral loads of three excluded animals must be included in supplemental data figure for review.
- 4) Is the historical data included in both Figure 1f and 1d, or only in 1f? It isn't clear from the text.
- 5) The T cell activation in ART Only animals increases significantly between d 217 p.i. and d 260 p.i. Do the authors know why? The large increase in the activation of memory CD4 T cells in the ART only group at this time significantly contributes to the statistical significance between groups.
- 6) Arrows in figure 1e representing IFN α administration should match color of arrows for IFN α administration in figure 1a.
- 7) The data represented in figure 1i (and all subsequent HLA-E restricted NK work) is difficult to understand. The methods and legend both say activity is measured by CD107a expression. However, somehow lysis is also measured and shown in the dotted line on the graph. How is the final timepoint for the cytokine treated group above 100? This wouldn't make sense for frequency CD107a or lysis.
- 8) Controls of NK cells stimulated with K562-E*0101 without SIV ENV peptide should be shown on the graph for both groups.
- 9) Why is no QVOA data shown for LN collected at day 35, prior to ART (figure 2g).
- 10) Do the authors have a hypothesis as to why the frequency of Ki-67+ T cells in PBMC is a better predictor of replication competent virus in LN compared to the frequency of Ki-67+ T cells in the LN itself?
- 11) Figure 4g: Authors should more clearly note in the text that there is no correlation between SIV ENV-specific NK cells and time to rebound in ATI at the d 13 post-ATI time point. Discuss.

Point-by-point Rebuttal

We thank the Reviewers for their thoughtful comments. The revised manuscript includes a large amount of new data that have been generated to address the main criticisms, including:

- More statistically stringent, re-visualized correlations demonstrating a clear role for IL-21/IFN α -induced, terminally-differentiated NK cells in facilitating lymphoid viral control during ART. These expanded analyses indicate that the NK cells of the cytokine-treated animals, in particular the *ex vivo* innate and HLA-E-restricted, SIV-Env-specific responses of terminally-differentiated (stage 3) NK cells, mediate viral control during ART. We further showed that the cytokine therapy-mediated enhancement of terminally-differentiated NK cell activity was maintained following ART interruption, in absence of IL-21 treatment and in the presence of viral replication, at least until the first two weeks after ATI, while the Env-specific NK cells responses were rapidly lost. Of note, the levels of less differentiated (stage 2) NK cells correlated with more rapid viral rebound.
- Flow cytometric analyses verifying that NK cell differentiation subsets are redistributed, but not expanded by cytokine therapy, which further suggests the presence of a “block” in NK cell maturation in pathogenic SIV infection. These analyses also demonstrate that therapy-induced terminally-differentiated NK cells are an important contributor to innate CD107a degranulation activity.
- Experimental detail describing how the NK cells were assayed either for their *ex vivo* innate activity or for SIV-Env-specific, HLA-E-restricted activity. The revised manuscript now includes representative flow cytometry plots, descriptions of co-culture conditions and calculations, and plots of the raw CD107a expression data.

We are therefore confident that thanks to the suggested revisions, the revised manuscript has been substantially improved through more detailed explanations of the approaches and additional analyses that strengthen the main conclusion of the study.

Reviewer 1.

Remarks to the Author: *Harper and Huot et al utilize a pathogenic model of SIV infection to determine the impact of IL-21 and IFN α cytokine therapy during ART on T cell activation, viral reservoirs and NK subsets, and following ATI. They identify changes in NK cell maturation states and cytolytic activity associated with cytokine therapy, which may be associated with viral reservoir size/rebound during ATI. The paper is well written, although the findings are occasionally overstated, and the figures are clear and well presented.*

Major Comment 1. *There are a few occasions where the findings are overstated and are not completely supported by the data. It would be helpful to rephrase. For example:*

- a. In the abstract the authors state that the “cytolytic activity of NKG2a/c low CD16+ NK cells... predicted viral rebound delay following ATI”. While the authors demonstrate a correlation between NK CD107a activity and days to rebound, this observational study is not powered for prediction.*
- b. Line 130-131 – “Altogether this implies that IL-21 regulates SIV persistence in the LN principally through an NK-mediated mechanism”. As above, correlations are insufficient to support the strength of this claim.*
- c. Line 197 - “this shows that NK cells have the capacity to mediate SIV control in tissues”. As above.*

Response: We agree that our data cannot definitely prove that the correlations between NK cell differentiation and viral control are causative. Some of these concerns can partially be mitigated with our companion manuscript by Huot et al., which has been resubmitted and is now approved-in-principle at *Nature Communications*, in which the authors demonstrate that NK cell maturation states found in SIV-infected African Green Monkeys (AGMs), but not RMs, exhibit unique anti-SIV-Env-specific properties *ex vivo*; indicative of a role in facilitating viral control. We will request that the editor forward the co-submission to reviewers for clarification. Furthermore, cytokine-treated RMs failed to display an enhancement of SIV-specific CD8⁺ T-cells responses, suggesting that an alternative mechanism is at play to reduce the content of replication competent virus *in vivo* (**Supplementary Fig. 5d,e**). As requested, the findings have been re-worded and tempered as stated in the passages below:

- a. “The frequency and cytolytic activity of terminally-differentiated NKG2a/c^{low}CD16⁺ NK cells correlated with a reduction of replication-competent SIV in LN during ART and viral rebound delay following analytical treatment interruption.” Note that we have removed comments on “prediction” throughout the manuscript.*
- b. “Of note, the content of LN replication competent virus, but not cell-associated SIV-DNA in tissues, displayed a unique positive correlation with HLA-E⁺ CD4⁺ T-cell levels and a negative correlation with Env-specific NK cell*

activity; suggesting that enhanced NK cell functionality is an important mechanism for IL-21-mediated reduction of the replication competent viral reservoirs.”

- c. “As terminally-differentiated NK cells were correlated with reductions in lymphoid replication competent virus during ART and the delay in viral rebound after ATI, these data support a role for SIV-Env-specific, HLA-E-restricted NK cells responses in controlling SIV in tissues.”

Major Comment 2. *One of the key interesting findings for is the cytotoxic activity of NK cell subsets with cytokine therapy. However, as currently presented this data is challenging to interpret.*

- a. *Could the authors explain why the data for this assay shown as log10 transformed? From the M&M (without the companion paper accessible) the final readout appears to be flow cytometric detection of CD107a.*
 b. *If this is indeed a flow-based assay, example plots are needed for interpretation.*
 c. *It would also be useful to add what the control for this assay is (e.g. unpulsed cell line?).*
 d. *Figure 4G – there is a solid grey box at CD107a+ NK vs d13 ATI, suggesting that the data does not exist for total NK. However, data is available for the NK subsets. Could the authors clarify why this data is missing please?*

Response:

- a. A detailed explanation of how the calculation is performed is given in response to reviewer 2, major comment #7. In brief, the % activity is calculated based on the %CD107a⁺ NK cells between conditions in which magnetically-isolated NK cells are cultured either alone, or co-cultured with MHC-I deficient K562 cells with induced HLA-E expression that are either loaded or unloaded with SIV-Env peptides. The calculated % activity was then log₁₀ transformed and treated as the ‘Env-specific activity’ (**Fig. 1i**), which is now explained in *Methods: Env-specific NK cell activity assays* and in the figure legend. If the data is treated as a percentage (below, at right) then the effects of a single positive outlier (RQb13; see response to reviewer 2, major comment #7) during therapy is greatly amplified resulting in an exceptionally high SEM, which still requires a log transformation of the y-axis to capture. A side-by-side comparison of the two modes of visualization is shown below. Note that for all longitudinal measures, the data has been re-visualized to show individual RMs color-coded by treatment overlaid against the mean and SEM to better show the data distribution.

- b. We have added representative flow plots of surface CD107a expression in NK cells from each of the co-culture conditions in **Supplementary Fig. 6d,e,f**. Furthermore, as the readout is solely based on CD107a expression by flow, we have rephrased the readout as “activity” as opposed to “lysis” or “cytotoxicity” throughout the manuscript.
- c. Internal controls for the Env-specific NK cell activity assay include isolated NK cells cultured alone to quantify background degranulation due to experimental conditions and isolated NK cells co-cultured with unloaded K562-E*0101 cells to quantify maximum degranulation capacity in response to the target cells. An explanation regarding the controls has been added to the *Methods: Env-specific NK cell activity assays*. We have also added plots of the raw CD107a data as part of the response to reviewer 2, major comment #8 (**Supplemental Fig. 6a,b,c**). Experimental validation of the specificity of the SIVmac-Env peptides is shown in the Huot et al. companion

manuscript. The assay also had external negative (VL9 peptide) and positive (HSP60 peptide) controls in which K562 cells expressing HLA-E*0101 were loaded with peptides based on their ability to block NK cell activity.

- d. This data was collected. The % CD107a⁺ NK cells in PBMCs has now been added to the d13 post-ATI correlations (**Fig. 4h**), and significantly associates with the delay in viral rebound as expected given a similar correlation with *ex vivo* innate responses in the stage 3 NK cells.

Major Comment 3. *Much of Figure 4 is built around a delayed rebound during ATI following cytokine therapy. However, as the authors note, this has been seen when pegIFN α alone was administered. This does not invalidate the potential role for NKs and subsets, but this section should be rephrased to highlight the pegIFN α caveat upfront (rather than lines 181-183), and to focus on the NK findings.*

Response: Reviewer 1 is correct, we cannot rule out that PEG-IFN α monotherapy following ATI would have also produced a delay in viral rebound. Notably, previous studies have demonstrated an inconsistent impact on PEG-IFN α on delaying viral rebound following ATI (Azzoni et al., *J Infect Dis*, 2013; Boue et al., *AIDS*, 2011). We have moved this caveat from the discussion to the results when introducing the ATI data in Fig. 4 and have re-phrased as follows (without references): “Of note, PEG-IFN α therapy has previously been suggested as able to delay viral rebound when initiated prior to ATI; whereas, in SIV-infected RMs, prior IL-21 monotherapy during ART is not”.

Major Comment 4. *In Figures 3C-F and 3G the authors nicely demonstrate the changes in NK subsets following cytokine therapy. However, the current analysis does not assess a change in frequency of these NK subsets, but a change in proportion with NK cells.*

- a. *It would be helpful to clarify this in the text (line 141).*
- b. *Modifying this analysis to ask what the percentage of these NK subsets is from CD45⁺ cells (rather than NK) may provide further insight to this finding. How do the proportional changes relate to the increase in total NKs? For example, when assessing a “block” between Stages 3 and 2, is the frequency of Stage 3 NK from CD45⁺ expanded, while Stage 2 is similar? Or is Stage 2 still lost? This could alter how this data is interpreted.*
- c. *Given the importance of the definition of these NK subsets, raw plots to see the gating strategy are needed here.*
- d. *Figure 3I – the increase in CD107a⁺ Stage 3 NK cells is really striking. However, the authors have not shown the data to support their claim (Line 158-160) that this population is solely responsible for CD107a⁺ expression by NKs. This would be shown by gating first on the CD107a⁺ NK cells, and then performing the subset analysis.*

Response:

- a. Correct, the values supplied for the subsets are given as frequency of NK cells (**Fig. 3c-f**); therefore, cytokine therapy-mediated effects should be viewed as a “redistribution” rather than an “expansion” as the reviewer suggests. We have rephrased the text as such and removed mentions of “expansion” elsewhere in the manuscript.
- b. We have now calculated the frequency of each NK cell subset of CD45⁺ lymphocytes (**Supplementary Fig. 9a,b,c,d**). Notably, cytokine treatment did not impact the distribution of the NK cell differentiation stages as a frequency of CD45⁺ lymphocytes. This observation is consistent with an absence of an impact on bulk NK cells (**Fig. 3a**) and confirms that cytokine therapy leads to a redistribution of NK cell differentiation subsets, rather than a *de novo* expansion, which reinforces the notion of a “block” at stage 2 as now described in the results.
- c. A representative gating strategy and longitudinal stains for the phenotypical discrimination of the NK cell subsets are now given as **Supplementary Figure 8**. This same definition and gating strategy for defining NK cells in non-human primates was recently described in Huot et al., *Front Immunol*, 2020.
- d. As requested, we have re-done the gating analysis in which we gated on CD107a⁺ NK cells and then gated for differentiation subsets (CD16 versus NKG2a/c). As now shown in **Supplementary Fig. 9e,f,g,h** we observe *ex vivo* innate degranulation activity in multiple NK cell differentiation subsets. Although there is a non-significant trend in which stage 3 NK cells from cytokine-treated RMs increasingly contribute to the pool of CD107a⁺ NK cells, there also exists a substantial contribution for less differentiated subsets. In the results, the description of this data has been re-worded and we have dropped the assertion that the stage 3 NK cells are the sole contributor to NK cell innate degranulation. However, as shown in **Fig.4h**, only the levels of stage 3 CD107a⁺ NK cells positively correlated with a delay in viral rebound.

Major Comment 5. *For the heatmaps (e.g. Figure 2H) all of the data (including the rho for non-significant p values) should be shown. Recent examples of alternative visualization approaches include Mathew et al Science 2020 and Moderbacher et al Cell 2020. Are these nominal p-values or has this been corrected for multiple comparisons? This may impact interpretation of this data.*

Response: All correlations in the submitted manuscript were previously performed as a two-sided Spearman rank correlation coefficient without a multiple comparison correction. As suggested we have re-analyzed the correlation data in SAS with the stepdown Bonferroni, Hochberg, and false discovery methods (FDR) for multiple comparison correction. All resulting correlation coefficients and statistics are now provided within the Source Data file (see tabs ‘Fig. 2h statistics’, ‘Fig. 3j statistics’, and ‘Fig. 4h statistics’). Notably, despite the small sample size, nearly all primary correlations remained significant following FDR correction with the exception of Env-specific NK cells responses on-ART being associated with viral rebound following ATI ($p=0.0597$; see also response to reviewer 2, major comment #11). Furthermore, all correlations and RNA-seq data have been re-plotted using ggplot2 (**Fig. 1f, 2h, 3j, 4d, 4h, and Supplementary Fig. 4**) allowing for the simultaneous visualization of the correlation coefficient or fold-change, the nominal p-value, and the adjusted p-value.

Minor Comment 6. *How were the gates for Figure 1E HLA-DR/CD38 determined? Is the same result seen when more stringent gates are applied (for example based on the naïve population)?*

Response: For activation ($\text{HLA-DR}^+\text{CD38}^+$) in memory CD4^+ T-cells and memory CD8^+ T-cells, the gates were set based on expression within naïve CD4^+ T-cells and naïve CD8^+ T-cells, respectively. This has now been noted in “*Methods: Flow cytometric analysis*”. As an example of the gating stringency, we show below longitudinal flow plots of HLA-DR versus CD38 in naïve CD4^+ T-cells in PBMCs, which are matched with the representative stains shown in **Fig. 1e**.

Minor Comment 7. *In Fig 1E, the control animal (RSp14) has few CD4 T cells at the initiation of ART in comparison to the cytokine-treated animal. Was the CD4 nadir equivalent between treatment arms?*

Response: CD4^+ T-cell counts at initiation of ART are shown for all animals in **Supplementary Table 1**. RSp14 indeed had the lowest count at d35 p.i. (184 CD4^+ T-cells per μL blood). However, there was no significant difference in the nadir CD4 count prior to ART between the control (361 ± 56.27 CD4^+ T-cells per μL) and cytokine-treated RMs (522.3 ± 60.85 CD4^+ T-cells per μL ; $p=0.1119$); nor did subsequent cytokine immunotherapy influence the frequency of PBMC CD4^+ T-cells (data not shown). Furthermore, differences in plasma viral loads were not significant different between control and cytokine-treated RMs at d35 p.i. by Mann-Whitney test ($p=0.8981$) or at any timepoint prior to ART initiation by 2-way ANOVA (all $p>0.9999$).

Minor Comment 8. *The language in lines Line 118-120 around Figure 2G is a little confusing when compared to Figure 1A, the M&M and the figure legend. Could the authors clarify what treatment was received by the controls in 2G: are these ART-only, or did they receive rIFN α ?*

Response: RMs shown for the QVOA in **Fig. 2g** are the same as those described in the study design in **Fig. 1a**. Throughout the manuscript, we have adjusted the phrasing to describe those animals receiving ART + IL-21 + rIFN α with PEG-IFN α following ATI as “cytokine-treated” and ART-only controls as “controls”. For the QVOA we were suggesting that the reduction in replication competent virus is primarily driven by the IL-21 therapy as levels are already reduced by d217 p.i.; whereas, subsequent rIFN α in cytokine-treated RMs or longer duration ART in controls (i.e. d217 to d374 p.i.) failed to produce a further significant reduction. We have re-worded the legend of Fig. 2g, the “*Methods: Quantitative viral*”

outgrowth assay (QVOA)", and the results to clarify. See also reviewer 2, comment #9 regarding a discussion of the QVOA timing.

Minor Comment 9. How sustained is the change in NK activity and subset distribution? Is it still present when all animals rebound at Day ~51 post-ATI? If samples are not be available or experiments are not possible, some discussion in the text would be interesting here.

Response: Our existing data demonstrates that at d13 post-ATI, by which 5/5 controls and 2/8 cytokine-treated RMs have rebounded, there is a clear enhancement in the frequency (Fig. 4e) and *ex vivo* innate degranulation activity (%CD107a⁺; Fig. 4f) of terminally-differentiated NKG2a/c^{low}CD16⁺ NK cells in cytokine-treated RMs as compared with controls. To partially address the concern regarding the durability of these NK cell responses, we have added the Env-specific NK cell activity at d13 post-ATI (Fig. 4g; see the response to reviewer 2, comment #11). These data show that Env-specific NK cell responses rapidly converge to values similar to those in early chronic infection (d35 p.i.) suggesting they are not sustained during viremic phases in contrast to the *ex vivo* innate activity of the total and terminally-differentiated NK cells. For reviewer evaluation, we also immunophenotyped PBMCs at d58 post-ATI (see below), by which time all RMs had undergone virologic rebound (Fig. 4c) and ISGs

were no longer stimulated by PEG-IFN α therapy (Fig. 4d). At d58 post-ATI we observed no sustained enhancement in the frequency of terminally-differentiated NK cells with values approaching those observed prior to ART initiation (Fig. 3f; cytokine-treated RMs: 14.17% \pm 9.566% at d58 ATI versus 6.569% \pm 1.739% at d35 p.i., p=0.0781). We have added a brief comment in the results highlighting that treatment maintained NK cell differentiation during the first 2 weeks following ATI, while the Env-specific activity was rapidly lost following ATI.

Reviewer 2.

Remarks to the Author: In their manuscript “IL-21 and IFN γ therapy rescues the maturation of terminally-differentiated NK cells and limit the SIV reservoir in ART-treated macaques”, Harper et al. follow the effects of sequential rIL-21 and rIFN γ treatment on the maturation of NK cells in the setting of pathogenic SIVmac239 infection. This is of particular interest because modulation of NK cell function is often overlooked when considering therapies to drive HIV-remission. The premise of the work is based on sound science and a wealth of previous knowledge gained from the study of NK cells during SIV/ART in rhesus macaques. The significance of the work is also high, since altering NK phenotypes in SIV-infected rhesus macaques to mimic that of non-pathogenic infected NHP (i.e. AGM) could lead to significant reductions in HIV morbidity and mortality. The results section is the only problematic section of the paper, and requires revisions and additional data to convince reviewer/author as to the veracity of the data.

Major Comment 1. Rectal biopsy needs to be appropriately abbreviated in the main text (line 91), and not just in the methods.

Response: The abbreviation for ‘rectal biopsies’ has been added in the results section (page 4; first time we quoted rectal biopsies) as requested.

Major Comment 2. How were PBMC time points chosen for different assays? For instance, figure 1d shows many data points for flow staining of PBMC, but then figure 1h shows a total of four timepoints of PBMC staining per cohort?

Response: For the *in vivo* study, we performed flow cytometry staining on fresh tissues with a panel designed to characterize T-cell responses for all of the collections listed in the study design (Fig. 1a; Supplementary Fig. 1). However, the immunophenotyping of NK cells, HLA-E⁺ CD4⁺ T-cells, and NKG2a/c⁺ CD8⁺ T-cells, in addition to *ex vivo* assays for NK cell activity, were performed on cryo-preserved PBMCs (see “Methods: Flow cytometric analysis”). In addition to performing time-matched viral reservoir characterizations and immunophenotyping for the correlations, our ability to perform the NK cell activity assays was limited by cellular yields given as they are performed with isolated NK cells. As such, we were restricted to performing these characterizations at experimental time points in which large volumes of blood were collected; immediately following an intervention cycle. As an implication, we cannot comment on whether NK cell

responses remain stable across the time periods between interventions. We have briefly commented on this limitation in the discussion.

Major Comment 3. *Viral loads of three excluded animals must be included in supplemental data figure for review.*

Response: Among the 9 cytokine-treated RMs, 172_10 experienced AIDS progression, starting very rapidly after infection, and at d66 p.i. was euthanized due to reaching the weight loss endpoint (-25% by nomogram). Since this animal was randomly assigned to one of the experimental arm, we elected to show his plasma viral load up until its humane endpoint necropsy (now shown in **Fig. 1b** and highlighted in red). Our laboratory has an active interest in determining features of a very rare subset of SIV_{mac239}-infected RMs able to control SIV after ATI (*Strongin et al. Virologic and Immunologic Features of Simian Immunodeficiency Virus Control Post-ART Interruption in Rhesus Macaques. J Virol, 2020*). Prior to ART initiation, RPK11 and RNA12 mimicked virologic and immunologic features of controllers; hence, they were removed from the analyses and not assigned to the cytokine-treated or the control group, but longitudinally followed as part of our studies aimed at characterizing post-treatment controllers. We have better described this point in the *Result* and *Method* sections of the revised manuscript. We feel that showing viral loads for those two animals, that were purposely not assigned to any of the experimental group, will be confusing and misleading for the readers; thus, we have provided the data below for this reviewer evaluation. However, we are fine to include them as supplemental figure if the reviewer strongly prefers to. Notably, if these post-treatment virological controllers were included as control RMs, as shown below, it does not change the finding that cytokine therapy significantly delayed viral rebound following ATI.

Major Comment 4. *Is the historical data included in both Figure 1f and 1d, or only in 1f? It isn't clear from the text.*

Response: While the activation data shown in **Fig. 1d** is from our longitudinal cohort (**Fig. 1a**), the data in **Fig. 1f** is from a different, internal unpublished cohort of RMs that received rIFN α after approximately 300 days of ART and in which RNA-seq analyses were performed. The main rationale for adding data from this cohort is to prove that the rIFN α used in the current study is indeed biologically active in RMs. We have removed the comment linking activation in memory CD4⁺ T-cells to the ISG upregulation as these are non-overlapping cohorts. Furthermore, data showing the longitudinal impact of ART on ISGs (**Supplementary Fig. 4**) are from a different external cohort (see end of “*Method: Study Design*”; Dunham, R.M., *et al. Pharmacologic inhibition of IDO blunts features of SIV-related chronic inflammation. in Conference on Retroviruses and Opportunistic Infections, Seattle, WA, 2017*). The text when presenting this data has been reworded to clarify.

Major Comment 5. *The T cell activation in ART Only animals increases significantly between d 217 p.i. and d 260 p.i. Do the authors know why? The large increase in the activation of memory CD4 T cells in the ART only group at this time significantly contributes to the statistical significance between groups.*

Response: This is an interesting question. While it is impossible with our set of data to directly address this question, one possibility is that the increased activation of memory CD4⁺ T-cells in the blood of control RMs between d217 and 260 p.i. is related to redistribution of activated cells from tissues to circulation. This can also reflect differences in levels of replication competent virus in tissues between the control and cytokine-treated RMs at that phase (**Fig. 2g**). Importantly, the most consistent and statistically significant difference in the term of immune activation among the two groups is at a

much earlier time point: following ART initiation during the first cycle of IL-21 (**Fig. 1d**). Those differences are not impacted by the increased in controls between d217 and d260 p.i.

Major Comment 6. Arrows in figure 1e representing IFN γ administration should match color of arrows for IFN α administration in figure 1a.

Response: As suggested, the colorations in the representative flow plot in **Fig. 1e** have been altered to match the study design and other plots.

Major Comment 7. The data represented in figure 1i (and all subsequent HLA-E restricted NK work) is difficult to understand. The methods and legend both say activity is measured by CD107a expression. However, somehow lysis is also measured and shown in the dotted line on the graph. How is the final timepoint for the cytokine treated group above 100? This wouldn't make sense for frequency CD107a or lysis.

Response: To clarify, the Env-specific NK cell activity shown in **Fig. 1i** is measured by CD107a surface expression and is calculated as follows, as previously described in Reeves, R.K., *et al.* Antigen-specific NK cell memory in rhesus macaques. *Nat Immunol* **16**, 927-932 (2015):

$$\log_{10} \left[100 \times \frac{\%CD107a_{NK+K562*E+ENV}^+ - \%CD107a_{NK}^+}{\%CD107a_{NK+K562*E}^+ - \%CD107a_{NK}^+} \right] = Env\ specific\ NK\ cell\ activity$$

In the presence of anti-CD107a-PE mAb, magnetically-isolated (NKG2a/c⁺) NK cells from cryo-preserved PBMCs were either cultured alone (i.e. %CD107a⁺_{NK}), co-cultured with K562-E*0101 cells without peptide (i.e. %CD107a⁺_{NK+K562*E}), or co-cultured with K562-E*0101 cells loaded with SIVmac_{239/251}-ENV peptides (i.e. %CD107a⁺_{NK+K562*E+ENV}). A breakdown of the formula is as follows:

- “Background” (i.e. %CD107a⁺_{NK}): This value corresponds to the capacity of NK cells to spontaneously degranulate in the absence of target cells. This variable is also dependent on experimental conditions, such as the quality of the samples (cryopreservation, speed of thaw, etc.), room temperature, media used for the culture, etc. According to the literature this value could change and therefore it is always measured.
- “Maximum degranulation” (i.e. %CD107a⁺_{NK+K562*E}): This value corresponds to the maximum degranulation potential of NK cells in response to targets (i.e. co-culture with MHC-I deficient K562 cells with induced expression of HLA-E*0101 without peptide). Related to the reviewer’s comment, an activity of greater than 100% can occur if there is lower CD107a expression for the maximum degranulation condition than in the peptide loaded condition (**Fig. 1i**). The “maximum degranulation” value is highly debated with some model assuming a value of 100% (Reeves *et al.*, *Nat Immunol*, 2015). This assumption would simplify the formula and change the activity as shown below; wherein, we still observe a similarly significant impact of cytokine therapy on the Env-specific NK cell activity. Although this approach eliminates the outliers and “fixes” data points with greater than 100% activity, we feel this assumption is incorrect for our set of data as experimentally we observe “maximum degranulation” values of much less than 100% (**Supplementary Fig. 6b**; see also the response to reviewer 1, major comment #2).

$$\log_{10} \left[100 \times \frac{\%CD107a_{NK+K562*E+ENV}^+ - \%CD107a_{NK}^+}{100 - \%CD107a_{NK}^+} \right] = Env\ specific\ NK\ cell\ activity$$

- “Peptide Activity” (i.e. %CD107a⁺_{NK+K562*E+ENV}): This value corresponds to the degranulation activity against the SIV-Env peptides displayed by the K562 cells with induced expression of HLA-E*0101.

An explanation of these control conditions and this calculation have been added to “*Methods: Env-specific NK cell activity assays*”.

To further clarify, this assay is designed to show that NK cells exhibit enhanced HLA-E-restricted, SIV-Env-specific responses. This assay is unrelated to the %CD107a⁺ plots shown in **Fig. 3i** and **4f**; wherein, CD107a expression was measured on NK cells without co-culture (see: “*Methods: Ex vivo innate NK cell activity assay*”). In the text and figures we have clarified this and referred to the latter as “*ex vivo innate activity*”.

Major Comment 8. *Controls of NK cells stimulated with K562-E*0101 without SIV ENV peptide should be shown on the graph for both groups.*

Response: We now show longitudinal plots in **Supplementary Fig. 6a,b,c** for the %CD107a⁺ of NK cells for all three culture conditions used to calculate the Env-specific NK cell activity as outlined in the response to reviewer 2, major comment #7. We have also added representative flow plots in response to reviewer 1, major comment #2b (**Supplementary Fig. 6d,e,f**).

Major Comment 9. *Why is no QVOA data shown for LN collected at day 35, prior to ART (figure 2g).*

Response: Historically, the QVOA was developed to specifically quantify the latent viral reservoir content during ART (Finzi et al., *Science*, 1997; Wong et al., *Science*, 1997; Laird et al., *PLoS Pathog*, 2013). In viremic patients the content of latent virus is approximately 100-fold higher than during ART and correlates with the plasma viral load (Blankson et al., *J Infect Dis*, 2000). Given a correlation with plasma viral loads, which for our animals ranged 10⁴ to x 10⁶ copies/mL at d35 p.i. (**Fig 1b,c; Supplementary Table 1**), this would make QVOA a less interesting analysis at this experimental time point as compared to phases in which all animals have comparable degree of virologic control. Of note, differences in plasma viral loads were not significant different between control and cytokine-treated RMs at d35 p.i. by Mann-Whitney test (p=0.8981) or at any timepoint prior to ART initiation by 2-way ANOVA (all p>0.9999).

Major Comment 10. *Do the authors have a hypothesis as to why the frequency of Ki-67+ T cells in PBMC is a better predictor of replication competent virus in LN compared to the frequency of Ki-67+ T cells in the LN itself?*

Response: This is another interesting and difficult to answer question. One possibility to explain proliferation in PBMCs being a better predictor of replication competent virus in LN is related to the anatomical distribution of Ki-67⁺ T-cells. For example, the frequency of proliferating T-cells in LN (**Supplementary Fig. 2h,k**) are not impacted by cytokine therapy and tend to be substantially lower than in PBMCs once on-ART (**Supplementary Fig. 2f,i**). As T-cell proliferation in LN is invariant, it cannot mathematically correlate with the content of replication competent virus in LN, which was significantly reduced by cytokine therapy (**Fig. 2g**). We speculate that given most proliferating effector T-cells egress the LN and migrate to blood (Reuter et al., *Cell Rep*, 2017; Pino et al., *PLoS Pathog*, 2019), this anatomical site is hypothetically most likely to show a varied distribution in proliferation responses with cytokine therapy and therefore most likely to correlate with other parameters. It is possible that PBMCs better reflect the overall immunological impact of the cytokine treatment than LN alone. Furthermore, it is plausible that IL-21 decreased virus not only in LN, but also in additional tissues that were not accessed or from which sufficient cells were not available to perform QVOA. For example, the cytokine therapy-induced reduction in RB cell-associated SIV-DNA content at d374 p.i. was nearly significant (**Fig. 2e**; p=0.0576), which was also negatively correlated with the frequency of stage 3 NK cells during ART and the delay in viral rebound at ATI. The levels in blood might reflect the overall, accumulative changes in tissues. We have now commented on this in the results when discussing the correlations in **Fig. 2h**.

Major Comment 11. Figure 4g: Authors should more clearly note in the text that there is no correlation between SIV ENV-specific NK cells and time to rebound in ATI at the d 13 post-ATI time point. Discuss.

Response: To address this comment, we have added in the revised manuscript a plot of the SIV-ENV-specific, HLA-E-restricted activity at d13 post-ATI (**Fig. 4g**). At d13 post-ATI, we still observe a clear enhancement in the frequency (**Fig. 4e**) and *ex vivo* innate activity (**Fig. 4f**) in the terminally-differentiated (NKG2a/c^{low}CD16⁺) NK cells in cytokine-treated RMs; however, the Env-specific response in control RMs is improved with the overall distribution mirroring what was observed in chronic infection prior to the intervention (**Fig. 1i**). As pointed out by the reviewer, the NK cell *ex vivo* innate and Env-specific activities on-ART are associated with the delay in viral rebound; whereas, following ATI this correlation only holds true for innate responses. It is plausible that NK cell Env-specific responses might be stimulated by viral replication or by high levels of antigen presentation, thus improving the responses in controls. It is also plausible that, despite cytokine treatment producing durable differences in NK cell differentiation when ART suppressed, the Env-specific responses in viremic phases are sensitive to exhaustion. Another possibility is that Env-specific NK cells responses are primarily stimulated by IL-21 therapy and wane over time. Here, for reviewer evaluation we have included a longitudinal plot of the Env-specific activity across all experimental time points (including d13 post-ATI, d415) and we have briefly addressed this point in the results. We also added a sentence at the end of the discussion to take this finding into more consideration: “As NK cell terminal differentiation was more strongly and consistently associated with viral control than the innate and HLA-E-restricted activity alone, these cells might exert additional antiviral properties, possibly including ADCC.”

REVIEWER COMMENTS

Reviewer #1 (Remarks to the Author):

The authors have comprehensively addressed my concerns in both the revised text/figures and the rebuttal.

Reviewer #2 (Remarks to the Author):

In this much improved manuscript, the authors show that NK cell differentiation and function is improved by cytokine treatment (rhIL-21 and rIFN α) during SIVmac infection and ART treatment.

The authors have successfully addressed all of my concerns and I think the data speaks for itself and is extremely important to publish given the implications.

REVIEWERS' COMMENTS

Reviewer #1 (Remarks to the Author):

The authors have comprehensively addressed my concerns in both the revised text/figures and the rebuttal.

Response: N/A

Reviewer #2 (Remarks to the Author):

In this much improved manuscript, the authors show that NK cell differentiation and function is improved by cytokine treatment (rhIL-21 and rIFN α) during SIVmac infection and ART treatment.

The authors have successfully addressed all of my concerns and I think the data speaks for itself and is extremely important to publish given the implications.

Response: N/A